# Dissected antiporter modules establish minimal proton-conduction elements of the respiratory complex I

Adel Beghiah[1], Patricia Saura [1,2], Sofia Badolato[1,2], Hyunho Kim[1], Johanna Zipf [1], Dirk Auman[1], Ana P. Gamiz-Hernandez [1], Johan Berg[1], Grant Kemp [1] & Ville R. I. Kaila [1] ✉

The respiratory Complex I is a highly intricate redox-driven proton pump that powers oxidative phosphorylation across all domains of life. Yet, despite major efforts in recent decades, its long-range energy transduction principles remain highly debated. We create here minimal proton-conducting membrane modules by engineering and dissecting the key elements of the bacterial Complex I. By combining biophysical, biochemical, and computational experiments, we show that the isolated antiporter-like modules of Complex I comprise all functional elements required for conducting protons across proteoliposome membranes. We find that the rate of proton conduction is controlled by conformational changes of buried ion-pairs that modulate the reaction barriers by electric field effects. The proton conduction is also modulated by bulky residues along the proton channels that are key for establishing a tightly coupled proton pumping machinery in Complex I. Our findings provide direct experimental evidence that the individual antiporter modules are responsible for the proton transport activity of Complex I. On a general level, our findings highlight electrostatic and conformational coupling mechanisms in the modular energy-transduction machinery of Complex I with distinct similarities to other enzymes.

The respiratory Complex I (NADH:ubiquinone oxidoreductase) is a redox-driven proton pump that reduces ubiquinone (Q) and pumps protons across the mitochondrial inner membrane or bacterial cytoplasmic membranes[1–5], powering ATP synthesis and active transport[6,7]. Complex I is a large (0.5–1 MDa) membrane-bound enzyme that initiates aerobic electron transport chains by oxidising nicotinamide adenine dinucleotide (NADH) and transferring the electrons along a 100 Å tunnelling wire composed of iron-sulphur (FeS) clusters to Q (Fig. 1a). The quinone reduction and subsequent dissociation of the quinol triggers proton pumping across the membrane domain[1–5,8,9], up to 200 Å away from the Q active site (Fig. 1a)[1–5,9–17]. Remarkably, this process is fully reversible, and Complex I can thus also operate in the

reverse mode, driving quinol oxidation and reverse electron transfer (RET), powered by a ΔpH-gradient across the membrane[18]. Despite several resolved structures[19–27], mutagenesis experiments[10–17], and computational studies[1,28–37], the long-range proton-coupled electron transfer (PCET) mechanism of Complex I is still not fully understood. In this regard, detailed mechanistic models have recently been proposed (*cf.* refs. 1,19,20,26,37,38), but with several conflicting interpretations of the directionality of the elementary charge transfer steps and the location of the proton pathways[1,19,20,30,37–39].

Previous studies suggest that the long-range proton pumping is catalysed by the antiporter-like subunits Nqo12, Nqo13, and Nqo14, together with Nqo7-11[1,23,30,31,37] that form bundles of transmembrane

[1]Department of Biochemistry and Biophysics, Stockholm University, 10691 Stockholm, Sweden. [2]These authors contributed equally: Patricia Saura, Sofia Badolato. ✉e-mail: ville.kaila@dbb.su.se

**Fig. 1 | Structure and function of complex I. a** Reduction of quinone (Q) triggers proton pumping in the membrane domain of Complex I. The subunit naming is based on *Thermus thermophilus* Complex I. **b** Closeup of antiporter-like subunits, Nqo12 (*left*, lateral helix not shown) and Nqo13 (*right*). Conserved residues along the hydrophilic axis are shown, and the trans-membrane broken helices TM7a/b and TM12a/b are highlighted in solid colour.

(TM) helices (TM4-8 and TM9-13). Molecular simulations suggest that these antiporter-like subunits establish the proton channels by symmetry-related elements[1,30,31,37] (Fig. 1a, b). Each antiporter module contains a hydrophilic axis with a Lys-His-Lys/Glu chain that provides an S-shaped water-mediated proton wire across the membrane[1,28–31], as recently also supported in part by high-resolution cryo-electron microscopic (cryoEM) structures (refs. 19,20,26,27,39, but see below). The broken helices TM7a/b and TM12a/b provide the additional flexibility required to control the opening of the proton channels[1,13,35,40]. Previous work[1,13,30,31,37] also found that the proton transfer barrier is modulated by the conformational state of a conserved Glu-Lys ion-pair, located at the interface of each antiporter-like subunit (Fig. 1b). More specifically, opening of the conserved ion-pair was found to lower the barrier of lateral proton transport within an antiporter sub-unit, and in turn, favour the conformational changes in the ion-pair of the neighbouring subunit, leading to the subsequent propagation of a protonation reaction (*cf.* refs. 1,30,37).

The long-range proton transport process was suggested[1] to take place by an electric wave that propagates in forward and reverse directions across the 200 Å wide membrane domain, in analogy to a Newton's cradle device[1,13,37] (with *back*-and-*forth* propagation of mechanical energy), and leading to the release and subsequent uptake of protons by conformational changes in the ion-pairs. This model was recently challenged[19,20] by an alternative proposal, where all protons are pumped by the terminal antiporter-like subunit (Nqo12 / ND5), and with the protons assumed to move horizontally across all subunits from the centre of the membrane domain both towards the quinone site and the terminal (ND5) edge[19,20]. A variation of this ND5-only model was also proposed where ND5 pumps all protons, but the three sub-units, ND2, ND4, ND5, together share the task of transferring two protons across the membrane[26]. These ND5-only models are based on the lack of experimentally resolved water molecules at the P-side of the membrane[19,20], although such locally dry patches are likely to establish

a tightly coupled proton pump by preventing a continuous contact across the membrane, which would lead to proton leaks[37]. The long-range energy transduction mechanism of Complex I has been of major interest for the last decades, with early proposals highlighting possible electrostatic and/or conformational effects involved in the process[4,10,23,41,42] (but *cf.* also refs. 1,5,20,21,29,30,37,38,43 for detailed molecular mechanisms).

Mutagenesis experiments[44] suggest that removal of the ND4-ND5 subunits leads to reduction of the proton pumping stoichiometry to one half, whilst recent experiments[16] found that certain substitutions in the proton channels resulted in apparent pumping stoichiometries of either 0 or 4 protons (per NADH), suggesting either complete or no uncoupling of proton pumping from the oxidoreduction activity, for reasons that still remain unknown (see Discussion). However, the direct biochemical characterisation of individual residues or even subunits involved in the proton transport process has so far been unsuccessful due to the tightly coupled proton- and electron transfer process, where mutations of key residues in the proton channels inhibit the Q-oxidoreduction activity, and vice versa[10–17]. Therefore, despite the detailed mechanistic proposals[1,19,20,37], there is currently no experimental validation of the principles underlying the proton transport in Complex I, which has thus remained one of the most controversial and debated questions in the field.

To derive a bottom-up understanding of the intricate proton pumping process, we design here minimal proton-conducting modules of Complex I that allow us to probe the molecular principles underlying the putative conductive states during turnover within the smallest proton-conducting unit of Complex I. Our constructs are built based on multi-scale simulations, and probed experimentally by bio-physical proton conduction experiments in proteoliposomes in combination with mutagenesis studies, with optimisation of genetic and molecular biological approaches that allowed the expression, pur-ification, and characterization of the engineered constructs (Supple-mentary Fig. 10, Supplementary Table 7). Our combined results show that the antiporter-like subunits Nqo13 (ND4) and Nqo12 (ND5) contain all functional motifs required to transport protons across membranes, and that the buried ion-pairs of the antiporter modules form key ele-ments that control the rate of proton conduction. On a general level, the established integrative approach provides a systematic basis to study the function of individual units in complex multi-subunit mem-brane complexes, as well as an approach to probe, *e.g.*, how disease-related mutations affect the individual proton transport reactions in Complex I.

## Results
### Principles of proton transport in antiporter modules
To rationally engineer and dissect the pumping machinery of Complex I, we first created atomistic molecular dynamics (MD) simulation models of the antiporter modules, Nqo13 and Nqo12^ΔTH (ΔTH - long transverse helix removed, see "Methods") from *Thermus thermophilus* Complex I (Fig. 2a, b) that form the respective penultimate and terminal modules in Complex I (Fig. 1a, Supplementary Fig. 1a). We embedded these protein models in a lipid membrane, surrounded by water molecules and ions (see "Methods"). During the microsecond molecular dynamics simulations, the isolated antiporter modules remain stable (Supplementary Fig. 1d–f), and dynamically closely resemble the analogous subunits within the intact Complex I simula-tions (Figs. 1, 2a, b, Supplementary Fig. 1d–f)[1,13,28–31]. Particularly, car-diolipin of the polar lipid model establishes tight contacts with the antiporter modules and seals them from the bulk phase (Supplemen-tary Fig. 4). In the Nqo13 construct, the water molecules establish an N-side input channel (from Lys282, via His218 and Asp228) to Lys235; a lateral proton pathway from the buried Lys235 (via His292) to Glu377; and an exit pathway from Glu377 vertically "downwards" with transient connections to the P-side bulk solvent (Fig. 2b, Supplementary Fig. 2).

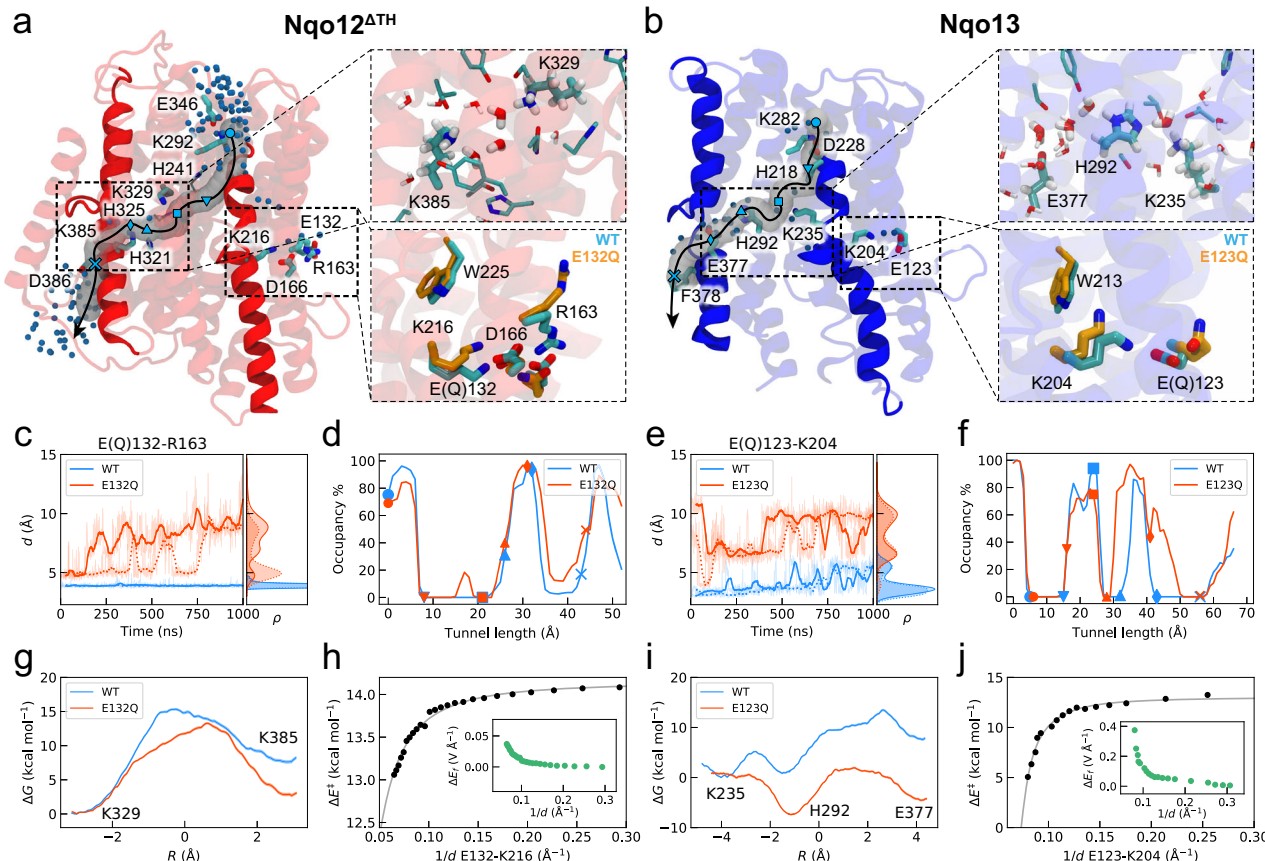

**Fig. 2 | Protonation pathways and modulation of proton transfer barriers.**
**a** Proton pathways from MD simulations of Nqo12^ΔTH. *Top inset*: MD snapshot showing water-mediated connectivity from Lys329 to Lys385 and conserved polar residues stabilising the proton pathways. *Bottom inset*: The Glu132/Arg163 - Asp166/Lys216 ion-pair conformation in the WT (cyan) and E132Q (orange) variants.
**b** Proton pathways from MD simulations of Nqo13. *Top inset*: MD snapshot of water-mediated connectivity from Lys235 to Glu377 via His292, and conserved polar residues stabilising the proton pathway. *Bottom inset*: the Glu123/Lys204 ion-pair conformation in WT (cyan) and E123Q (orange) variants. **c**, **e** Ion-pair dynamics and **d**, **f** hydration level along the proton pathways in the WT and E132Q/E123Q variants of Nqo12^ΔTH (**c**, **d**) and Nqo13 (**e**, **f**); symbols in (**d**, **f**) refer to the pathway region indicated in panel (**a**) and (**b**), respectively. **g**, **i** Free energy profiles of proton transfer along the proton channel in the WT (blue) and E132Q (E123Q, orange) variants of (**g**) Nqo12^ΔTH and (**i**) Nqo13. **h**, **j** Modulation of proton transfer barrier with ion-pair distance based on QM/MM calculations (black dots) and a fit from a theoretical model (see Supplementary Information). *Inset:* dissociation of the ion-pair induces an electric field (green dots) along the proton pathways. Data are provided in the Source Data file.

The N- and P-side connections are gated by the conformational state of the bulky non-polar residues (N-side: Leu214, Phe326; P-side: Phe378, Supplementary Fig. 1h, i, *cf.* also refs. [16],[30]), that together with subtle conformational changes in the broken helices TM7a/b (N) and TM12a/b (P) modulate the hydration level of the channels (Fig. 2f, Supplementary Fig. 2). Several of these potential gating residues are conserved (Supplementary Fig. 18), with conformational changes observed upon proton transfer between the central residues that could regulate the proton uptake and release from the bulk (see Supplementary Fig. 1g, h and Supplementary Discussion).

Analogous S-shaped water-mediated proton pathways also form during the MD simulations of the Nqo12^ΔTH construct, with an N-side access pathway (from Glu346, via Lys292 to Lys329); a lateral pathway (from Lys329, via 2-4 water molecules to Lys385); and further a P-side pathway via Asp386 to the P-side bulk (*cf.* also refs. [1],[19],[20],[28],[30], Fig. 2a, d, Supplementary Fig. 3). Several hydrophilic residues (N-side: Tyr94, Thr23, Ser237, His241; lateral pathway: His321, His325, Gln302, Tyr305) stabilise these proton wires and could support the proton transport (see also Supplementary Fig. 2).

The buried ion-pairs are located ~14/22 Å from the main proton pathways in Nqo13/12 and electrostatically modulate the barrier for proton transport along the main channel (see below, ref. [1], and Supplementary Figs. 5, 8). In Nqo13, the ion-pair (Glu123-Lys204) samples a

closed conformation in the MD simulations (Fig. 2b, e, Supplementary Fig. 5), with transient dissociation events that are stabilised by a cation-π interaction with a nearby tryptophan residue (Trp213, Supplementary Fig. 2h, *cf.* also ref. [30]). In Nqo12^ΔTH, the ion-pair (Lys216/Asp166; Arg163/Glu132) is located around 20 Å from the central lysine residue (Lys329), and also undergoes transient dissociation events that could modulate the proton transfer barriers, despite the larger distance as compared to Nqo13 (Fig. 2c, Supplementary Figs. 3, 5).

To estimate the proton transfer energetics along the water-mediated proton wires, we next employed hybrid quantum/classical (QM/MM) free energy calculations performed based on the microsecond MD simulations that allowed us to capture the energetics for the proton transfer along the lateral pathways (Fig. 2g, i, Supplementary Fig. 6). These lateral proton pathways were the main focus here due to their proximity of the ion-pairs and the central proton transfer elements (but see below). For Nqo13 structures with a closed ion-pair conformation, we obtain a proton transfer barrier between Lys235 and Glu377 of $\Delta G^{\ddagger}$ ~ 14 kcal mol⁻¹ and an overall $\Delta G$ ~ + 7 kcal mol⁻¹ (Fig. 2i), whilst dissociation of the (Lys204-Glu123) ion-pair drastically lowers the proton transfer barrier by electrostatic effects (Fig. 2j, see below). Although the complete free energy exploration of the *ca.* 50 Å long proton transfer pathway is not technically feasible with the current methods, we note that the proton uptake to the middle lysine is not

rate-limiting for the overall process, with a rather low reaction barrier ($\Delta G^{\ddagger} < 5$ kcal mol$^{-1}$, see "Methods", Supplementary Fig. 6).

In order to induce dissociation of the ion-pair, mimicking a conductive state along the wave propagation model[1,13,37], we replaced Glu123 (Glu132 in Nqo12) by a glutamine residue in the simulation models (Fig. 2b). Indeed, during the MD simulations of the Nqo13-E123Q variant, we observe a drastic increase in the dissociation of the Gln123-Lys204 distance (Fig. 2e), which in turn decreases the Lys204-Lys235 distance (Supplementary Fig. 5b), whilst the water array between Lys235 and Glu377 is retained (Supplementary Fig. 2b, f). MD simulations of the Nqo12$^{\Delta TH}$-E132Q show similar dissociation of the ion-pair (Fig. 2c) coupled to a subtle decrease in the Lys216-Lys329 distance (Supplementary Fig. 5g), but also a change in the overall hydration level of the lateral and N-side proton input channel (Fig. 2d, Supplementary Fig. 3). These findings suggest that the conformational state of ion-pair could regulate the proton uptake and release by wetting transitions, and vice versa (see Discussion).

In order to block the possible proton pathways, we replaced the middle lysine in Nqo13 and terminal lysine in Nqo12 with Met or Ile residues (K235M[13] and K385I[12]), due to their similar volume but non-polar character. Our MD simulations of these constructs suggest that these substitutions could indeed lead to a partial sealing of the proton pathway (Supplementary Figs. 2c–f, 3c, d).

Consistent with the electrostatic modulation of the reaction barrier, our QM/MM free energy calculations suggest that the E123Q variant of Nqo13 both kinetically ($\Delta G^{\ddagger} \sim 6$ kcal mol$^{-1}$) and thermodynamically ($\Delta G \sim -5$ kcal mol$^{-1}$) strongly favours the lateral proton transfer along the proton pathways (Fig. 2i). We find that the ion-pair opening induces an electric field vector along the Lys235 → Glu377 direction that could provide a driving force for the barrier modulation (Supplementary Fig. 8b, d). This is also supported by our analytical model (Fig. 2j, Supplementary Fig. 7, Supplementary Methods), predicting a similar dependence on the barrier modulation when accounting for the geometry of the antiporter architecture.

In Nqo12$^{\Delta TH}$, the conformation of the double ion-pair also modulates the free energy barrier and thermodynamic driving force for the proton transfer along the lateral pathway (Fig. 2g, closed state: $\Delta G^{\ddagger} \sim 15$ kcal mol$^{-1}$, $\Delta G \sim 7$ kcal mol$^{-1}$; open state $\Delta G^{\ddagger} \sim 13$ kcal mol$^{-1}$, $\Delta G \sim 2$ kcal mol$^{-1}$), despite the larger distance between the motifs as compared to Nqo13 (Supplementary Fig. 5). The barrier modulation in Nqo12$^{\Delta TH}$ also follows the electrical tuning model (Fig. 2h, Supplementary Fig. 8a, c). Taken together, our findings support that electric field effects modulate the protonation dynamics in the Nqo proton channels.

## Antiporter modules conduct protons in proteoliposomes

To experimentally probe the proton conduction properties of the dissected antiporter-like modules, we next created His-tagged variants of the Nqo13 and Nqo12 constructs fused with a superfolder GFP (sfGFP)[45], which allowed us to monitor the protein stability and localisation by the GFP fluorescence. We expressed the constructs in *E. coli*, and purified the detergent-solubilised constructs on a Ni-NTA column and by size-exclusion chromatography (SEC), followed by reconstitution of the purified proteins into proteoliposomes comprising *E. coli* polar lipids (see "Methods", Supplementary Fig. 10), to mimic the native bacterial membrane.

Initial expression of the isolated antiporter modules resulted in rather poor yields, but we could optimise the expression and purification procedure by systematically testing different constructs and expression vectors (Supplementary Fig. 10a–c, Supplementary Table 7). In this regard, addition of a C-terminal sfGFP increases the protein stability and yields, whilst introduction of a GSAGS linker between the C-terminus and the sfGFP-TEV cleavage site enhanced the accessibility of the His-tag for the Ni-NTA for the Nqo12$^{\Delta TH}$ construct

(Supplementary Fig. 10d–f, h, i, Supplementary Table 6, 7). Our GFP fusion-based fluorescence-detection/size-exclusion chromatography (FSEC)[45] further supported that all expressed and purified protein constructs were well-folded (Supplementary Fig. 10j, k).

To probe the proton conduction properties of the antiporter modules, we next monitored the proton transfer across proteoliposome membranes using pyranine (8-hydroxypyrene-1,3,6-trisulfonic acid, HPTS)−a fluorescent dye that is quenched as a response of a pH change across the proteoliposome membranes[46] (see "Methods", Fig. 3a, Supplementary Fig. 11a, Supplementary Fig. 12a), by optimising the liposome size and Nqo-*to*-liposome ratio (Supplementary Fig. 13a). As we have removed the energy-transducing quinone-oxidoreductase machinery of Complex I, the isolated antiporter modules can thermodynamically transfer the protons across the proteoliposome membranes, only when driven by an external proton motive force ($\Delta\mu_{H^+}$). Atto-labelling of the His-tag[47] prior and after proteoliposome solubilisation with DDM further suggest that the modules align with a 75% N-side inward orientation in the proteoliposomes (see "Methods", Supplementary Table 2, Supplementary Fig. 10g). Moreover, in-gel fluorescence suggested that all constructs showed the same level of antiporter incorporation into the liposomes (Supplementary Fig. 10o).

The proton conduction across the proteoliposome membranes was triggered by addition of acid or base that creates a proton gradient ($\Delta pH$) across the proteoliposome membrane. The build-up of an electrical component ($\Delta\psi$) of the $\Delta\mu_{H^+}$ that could block the proton conduction, was dissipated by addition of valinomycin, which allows K$^+$ ions to equilibrate across the membrane. Although the focus here is on the $\Delta pH$-driven proton transport, we note that the antiporter modules also support a $\Delta\psi$-driven proton transport, with the valinomycin sensitivity suggesting that the transport could be electrogenic (see "Methods", Supplementary Fig. 13b, c, cf. also refs. [48,49]). The experiments were started by setting the inside pH of the proteoliposomes to 7.2, based on the p$K_a$ or pyranine at 25 °C (see Supplementary Fig. 12a), and leading to a maximal change in the HPTS fluorescence with $\Delta pH$ based on calibration measurements (Supplementary Fig. 12a). The proton ionophore nigericin was added at the end of the measurement to equilibrate the proton concentration between the two compartments. Moreover, as controls, we also applied the same approach to study the proton conduction in empty liposomes as well as liposomes reconstituted with aquaporin (AqpZ), a water channel of similar size (80 kDa)[50–52] as the isolated antiporter modules.

To induce specific acidification of the inside of the liposomes, we performed experiments by adding potassium acetate that diffuses in the neutral form (CH$_3$COOH) across the membrane, but becomes trapped on the inside of the (proteo)liposomes in its charged form (CH$_3$COO$^-$) with p$K_a \sim 4.8$, and induces a $\Delta pH$ (Fig. 3a, Supplementary Fig. 12a, b). Addition of 10 mM potassium acetate to the empty liposomes and the AqpZ-proteoliposomes results in a rapid drop of the interior pH by 0.3 pH unit, followed by a slower alkalinisation possibly due to a slow non-specific leak reaction (Fig. 3b, c, Supplementary Table 1). We also note that some pyranine could have remained bound to the outside membrane surface, despite purification on the PD10 column. As the protons cannot freely diffuse across the membranes, the system establishes a Donnan equilibrium between the CH$_3$COO$^-$ (bound by the equilibrium with the CH$_3$COOH form), Cl$^-$, as well as the K$^+$ ions (diffusing via valinomycin) that is dissipated upon addition of nigericin (Fig. 3b, c). In stark contrast to the empty liposomes and AqpZ, addition of potassium acetate to the Nqo13 and Nqo12 proteoliposomes resulted in rapid Nqo-mediated proton transport (<600 ms, see "Methods") across the proteoliposomes membrane (CH$_3$COOH$_{in}$ → CH$_3$COO$^-_{in}$ + H$^+_{in}$ → H$^+_{out}$, Fig. 3a) as indicated by the fast pH increase and a shifted equilibrium between the CH$_3$COO$^-$ / CH$_3$COOH forms by dissipation of the Donnan potential (Fig. 3a–c). In this regard, the steady-state $\Delta pH$ level reaches 7.1–7.12 for the WT

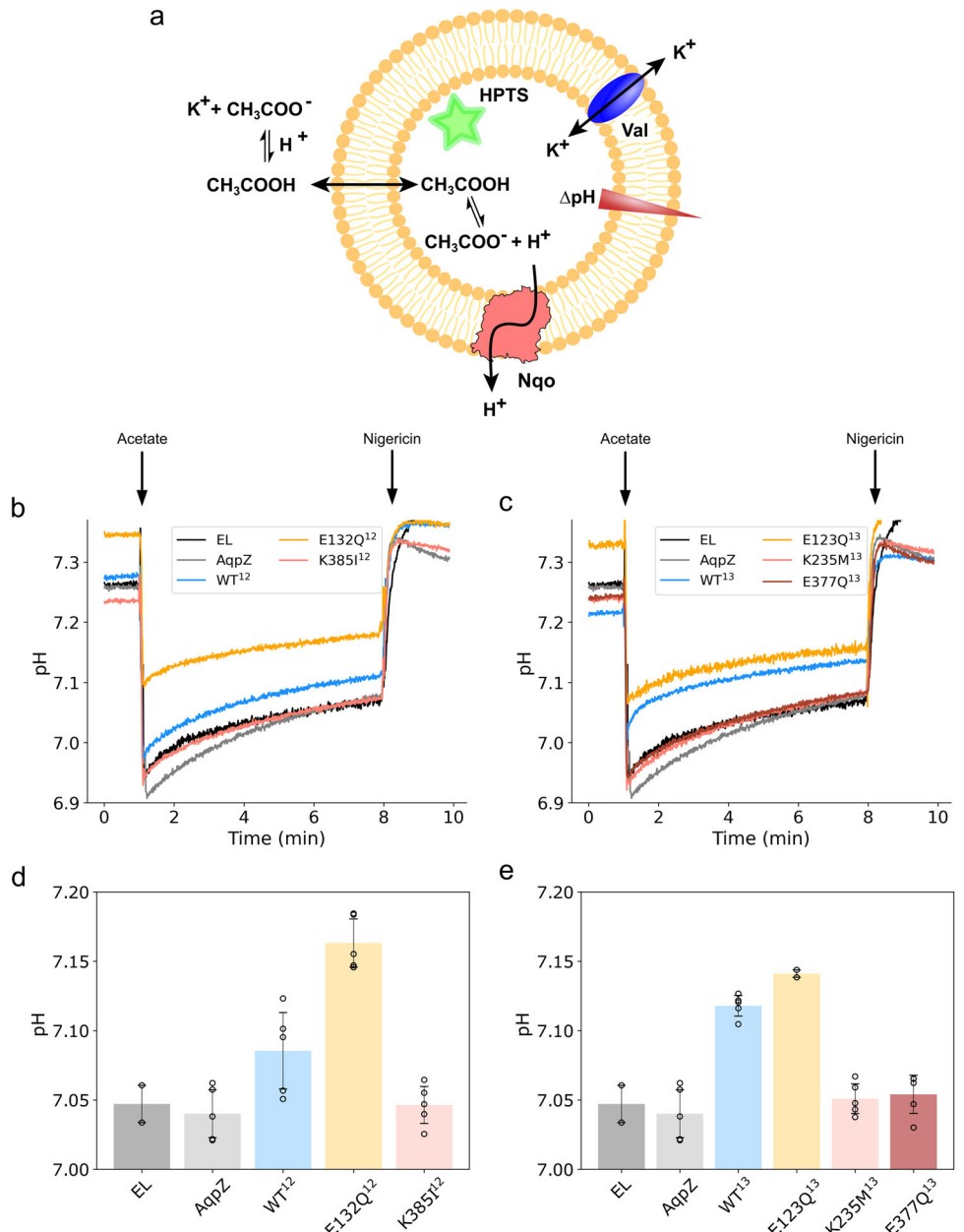

**Fig. 3 | Biophysical characterisation of proton conduction in dissected antiporter-like modules induced by addition of weak acid. a** Proteoliposome assay for probing the proton conduction kinetics in the dissected antiporter-like subunits with pyranine (HPTS) by addition of K⁺CH₃COO⁻. Addition of 10 mM acetate to (**b**) Nqo12$^{\Delta TH}$- and (**c**) Nqo13- proteoliposomes induced conduction of protons across the membrane. The ionophore nigericin was added to dissipate the generated ΔpH. The final steady-state ΔpH level before addition of nigericin is shown for the (**d**) Nqo12$^{\Delta TH}$ and (**e**) Nqo13 constructs. The protonation conduction rate is sensitive to substitution of the conserved ion-pair and residues along the proton pathway. The data is compared to proton conduction assayed with empty liposomes (EL, $n = 3$) and aquaporin (AqpZ) reconstituted proteoliposome. Note that the initial pH gradient pre-equilibrates upon mixing the proteoliposomes in the buffer, leading to a higher initial baseline for the fast-conducting constructs (E123Q[13], $n = 3$/ E132Q[12]), while the final steady-state pH (**d**, **e**), obtained prior to addition of nigericin, is independent of the baseline. Data shown are derived from independent experiments where $n = 6$ (mean ± SD) unless specified differently. See Supplementary Fig. 11 for addition of weak base, and Supplementary Fig. 13b, c for ΔpH and Δψ-mediated proton transport. Data are provided in the Source Data file.

Nqo13 and Nqo12$^{\Delta TH}$ relative to the empty liposomes/AqpZ (ΔpH<7.04), while the substitution of the ion-pair glutamate at the position Glu123 (E123Q-Nqo13) or Glu132 (E132Q-Nqo12) strongly increases the proton conduction rates (Fig. 3b–e, Supplementary Table 1), and leading to a further shift in the steady-state ΔpH level to a pH of around 7.15, consistent with our computational predictions of the lowered proton transfer barrier (Fig. 2c–j). Moreover, our experiments show that proton conduction can also be blocked by substitutions of the key residues along the proton wire in the studied conditions, leading to the same steady-state ΔpH level and proton conduction kinetics as for the empty liposomes. In this regard, we find that the removal of the terminal Lys385 (K385I) in Nqo12$^{\Delta TH}$, or replacement of the middle lysine (K235M) or the terminal glutamate (E377Q) in Nqo13 (Fig. 3b–e) also blocks the proton conduction, suggesting that the residues are important for establishing a tightly gated proton transport machinery in Complex I. These differences are highly unlikely to arise from protein aggregation or unfolding effects, as shown by size-exclusion chromatography and FSEC-GFP profiles (Supplementary Fig. 10d, j, k) and in-gel fluorescence (Supplementary Fig. 10o). Moreover, aggregated protein constructs would not allow us

to both enhance and block the proton conduction to the control level by introducing rational mutations (see below). We also find that addition of methylamine ($CH_3NH_3^+ / CH_3NH_2$), in which the direction of proton transport is reversed relative to the acetate-induced transport, resulted in a pH increase that also competes with the rate of the Nqo-mediated proton uptake rate (Supplementary Fig. 11).

To overcome the background arising from orientational effects of the Nqos in liposomes, association of HPTS to both sides of the membrane (see above), as well as possible leak reactions, we next co-reconstituted ATP synthase ($F_1F_O$-ATP synthase from *E. coli*) together with the different Nqo constructs, and probed their proton conduction properties relative to AqpZ and empty liposomes (see "Methods"). To this end, we initiated the proton pumping by addition of 0.2 mM ATP on the outside of the proteoliposomes, generating an ATPase-driven *pmf* across the membrane (Fig. 4a). As this ΔpH gradient can transiently become rather large, we followed the proton transport by the fluorescence quenching of 9-amino-6-chloro-2-methoxyacridine (ACMA), which has a broader pH range relative to pyranine, and commonly used for characterising the *pmf* linked to ATP synthase[53]. We find that addition of 0.2 mM ATP to the empty liposomes does not induce fluorescence quenching of ACMA, providing a stable background signal. However, when ATP is added to the proteoliposomes co-reconstituted with either AqpZ and ATP synthase, or with ATP synthase alone, we observe a rapid generation of a ΔpH across the membranes ($k_1(F_1F_O)$=5.56 min$^{-1}$; $k_1(AqpZ+F_1F_O)$=4.42 min$^{-1}$), reaching a stable plateau indicative of non-leaking vesicles, and reaching an overall quench level of 90% (Fig. 4d–g). The proton transport depends on the concentration of ATP (Supplementary Fig. 14a), and the ΔpH can be dissipated by addition of nigericin, which restores the fluorescence signal of ACMA (Fig. 4d, f). We find that liposomes with ATP synthase and AqpZ or ATP synthase alone are quenched to the same extent, thus reinforcing that AqpZ does not conduct protons. In contrast, for the Nqo12 or Nqo13 constructs, the ATPase-driven acidification is significantly slower ($k_1$ = 2.53 min$^{-1}$ and $k_1$ = 2.36 min$^{-1}$, Supplementary Table 1) and reaches around 30% steady state quenching level for both Nqo13 and Nqo12, showing that the Nqo constructs conduct protons. Moreover, introduction of the ion-pair substitution (E132Q[12] and E123Q[13]) increases the proton transport rate, with the Nqo-mediated proton transport almost completely out-competing the proton transport rate of ATP synthase for E132Q[12] (30% quench, Fig. 4b, d, e) and E123Q[13] (50% quench, Fig. 4c, f, g), whereas the substitution of the middle or terminal lysine residue (K235M[13], K385I[12]) leads to a partial or full block of the ATPase-induced proton transport, consistently with our acid-induced proton conduction experiments (Fig. 3), but with the subtle differences resulting from the different PMF, pH conditions, as well as buffering capacities used in the experiments (see Methods). The rate of proton transport in the *E. coli* ATP synthase at 120 mV is around 2400 H$^+$ s$^{-1}$ in the hydrolysis mode[48], suggesting that the Nqo constructs can catalyse the elementary proton transport on the sub-millisecond timescales.

Taken together, our proteoliposome experiments show that Nqo13 and Nqo12 comprise all necessary elements required to conduct proton across membranes upon stimulation with an external ΔpH gradient, and that the conduction rate is modulated by the buried ion-pair element, consistent with an electrostatic tuning model that also underlies the electric wave propagation model[1].

## Discussion

By combining biophysical proton conduction assays with mutagenesis experiments and multi-scale molecular simulations, we show here that the isolated antiporter modules of Complex I conduct protons across the membrane when stimulated with an external ΔpH gradient. Importantly, as proton conduction is observed for both Nqo12 (ND5) and Nqo13 (ND4), our data challenges the recent ND5-only pumping model[19,20]. In this regard, our data support that the proton pathways as

well as barrier tuning principles in the antiporter modules follow the same overall principles as in the native Complex I, despite also some key differences (Fig. 5, see below). Similar to the intact Complex I, the antiporter modules showed an N-side proton input channel at the broken helix TM7a/b; a lateral proton pathway connecting the middle lysine with a terminal charged residue (Glu or Lys); and a proton release pathway next to the broken helix TM12b leading from the P-side bulk (Fig. 2). However, the antiporter modules lack the native inter-subunit contacts that affect p$K_a$ values of the charged residues at the Nqo12-13 interface, downshifting the p$K_a$ of Glu377 and upshifting the p$K_a$ of Lys235 of Nqo13 (Supplementary Fig. 16, Supplementary Table 3), with recent experiments also supporting electrostatic tuning of these sites[54], and with the mutations in ion-pairs (*e.g.* E123Q[13]) inducing conformational changes and p$K_a$ shifts as also supported by our constant pH-MD simulations (Supplementary Fig. 16c). Moreover, the proton transport to the P-side takes place in Nqo13 at the lipid-protein interface, whereas the intact Complex I utilises the interface between the Nqo12 and Nqo13 subunits for the proton release[30,31] (Fig. 5). Nqo12 lacks a neighbouring subunit on one side, and we find overall smaller p$K_a$ shifts upon isolation of Nqo12 (Supplementary Fig. 16, Supplementary Table 3), suggesting that its protonation dynamics could resemble that of the intact Complex I. In this regard, our constructs also show similar hydration at the protein-lipid interface as compared to the protein-protein interface of neighbouring subunits in simulations of the intact Complex I[30] (Supplementary Fig. 19), and no signs of unfolding or aggregation (Supplementary Figs. 10) that could lead to artificial proton leaks.

To test the effect of a possible Nqo12-Nqo13 interactions, we attempted both co-expression and co-reconstitution of Nqo12 and Nqo13. Co-reconstitution of Nqo12 and Nqo13 in proteoliposomes resulted in some subunit dimerisation (Supplementary Fig. 17c). However, based on analysis of blue native gels, this oligomerisation could originate from self-assembled modules (Supplementary Fig. 17c), whereas co-expression of Nqo12 and Nqo13 resulted in protein aggregation (Supplementary Fig. 17a, b). Taken together, these findings suggest that the design of a specific Nqo12-13 dimer requires re-engineering of the transverse helix to clamp the subunits together.

Mutation of the buried ion-pairs in the intact Complex I results in the complete inhibition of the proton pumping as well as coupled oxidoreductase activity[12,14,15], thus hampering systematic studies of how this site modulates the proton transport kinetics. While these studies show that the coupled activity is diminished, they do not allow to determine how the mutations perturb the individual proton transfer reactions within the membrane domain. Here we decoupled the oxidoreduction activity from proton transport, allowing us to resolve how the ion-pair dynamics and individual residue substitutions control the rate of trans-membrane proton transfer. In this regard, we found that opening of the ion-pair, mimicked by the E123Q/E132 substitutions in Nqo13/12, creates an orientated electric field along the lateral channel that lowers the free energy barrier for lateral proton transfer from Lys235 via His292 to Glu377 (Lys329 to Lys385/Asp386 in Nqo12), from where the proton can be conducted to the bulk side of the membrane (Fig. 2). The ion-pair conformation also affects the hydration level of the central channel by inducing subtle conformational changes in the residues located on TM7a/b helix (Supplementary Figs. 2g, 3e). Such allosteric coupling elements could allow for a crosstalk between the proton uptake and release, by linking the ion-pair motion to protonation changes in the neighbouring antiporter-like subunit. Our MD simulations also suggested that mutation of residues in the proton channels perturbs the overall hydration state of the proton pathways relative to the wild type modules. Introduction of non-polar residues resulted in conformational changes along the proton channel that enhance the connectivity to the bulk solution (Supplementary Figs. 2, 3), consistent with the increased proton conduction rate (Figs. 3, 4). We expect that carefully balanced interactions along the proton

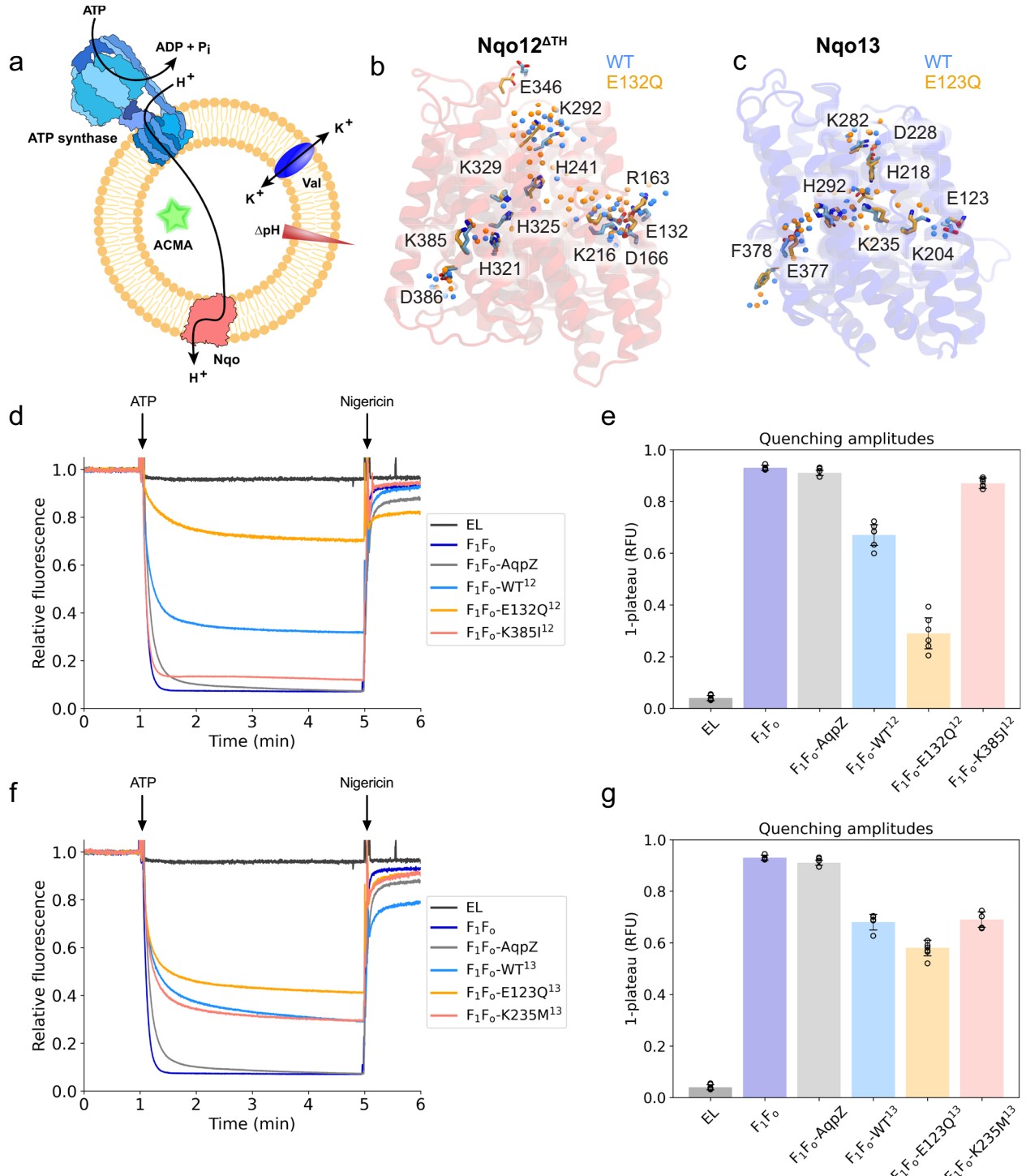

**Fig. 4 | Biophysical characterisation of proton conduction properties in dissected antiporter-like modules co-reconstituted with $F_1F_o$-ATP synthase.**
**a** Proteoliposome assay for probing the proton conduction kinetics in the dissected antiporter-like subunits co-reconstituted with ATP synthase, monitored by fluorescence quenching of ACMA. Addition of 0.2 mM ATP generates an ATPase-driven $\Delta pH$ across the proteoliposome membrane, which competes with the (**d**) Nqo12$^{\Delta TH}$- and (**f**) Nqo13-mediated proton transport. **b**, **c** Structure of the proton pathways from MD simulations of the WT (blue) and ion-pair mutants (orange) for (**b**) Nqo12$^{\Delta TH}$ and (**c**) Nqo13. Sidechains of conserved residues are shown as sticks, water molecules as spheres. Relative ACMA quenching amplitudes for co-reconstituted ATP synthase with (**e**) Nqo12$^{\Delta TH}$ and (**g**) Nqo13 constructs. Data shown are derived from independent experiments where $n = 6$ (mean ± SD), except $F_1F_o$-K235M$^{13}$ where $n = 5$. Data are provided in the Source Data file.

pathways are central for establishing the tightly-gated long-range proton pumping in Complex I.

In the native Complex I, the conformational state of the ion-pairs at the interface of the antiporter-like subunits could be regulated by the protonation state of the terminal residue of the previous subunit or by transiently sharing the pumped proton with the neighbouring subunit. Although the E132/123Q variants were introduced here to mimic an "open" ion-pair conformation, these substitutions could also be considered as models of a state, where neighbouring antiporter modules partially or completely exchange a proton. Indeed, Röpke

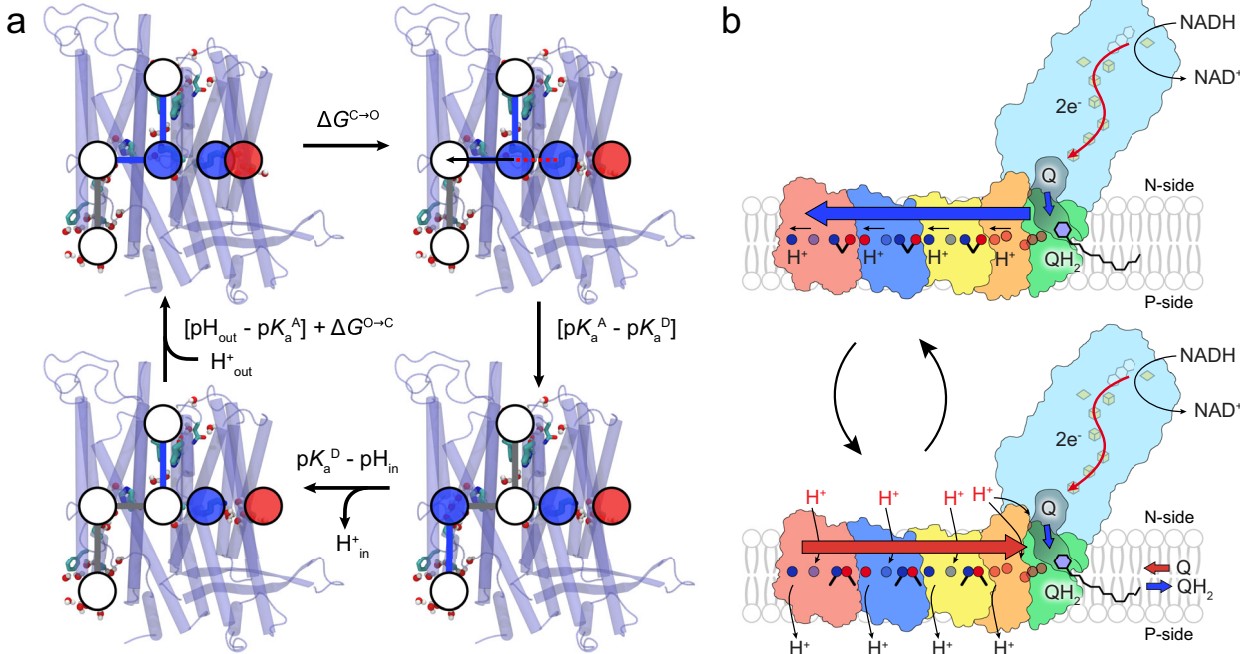

**Fig. 5 | Mechanistic model of proton transport in the antiporter modules.**
Schematic model of (**a**) antiporter-mediated proton transport. Blue/grey lines schematically show conneted/disconnected proton wires, and red dotted line an electrostatic repulsion. Blue/red/white circles – putative positive/negative/neutral protonation states. $pK_a^A$ – $pK_a$ of N-side proton acceptor; $pK_a^D$ – $pK_a$ of P-side proton donor; $\Delta G^{O \to C}$ – free energy closing ion pair; $pH_{out}$ – pH of N-side; $pH_{in}$ – pH of P-side. **b** Comparison of a proton pumping model in the intact Complex I.

et al.[31] found that the protonation of the ion-pair carboxylates at the Nqo13-12, Nqo13-14, Nqo14-10/11 interfaces induced conformational changes in the mouse Complex I that lead to opening of the ion-pairs and lowering of the lateral proton transport barrier. However, they also observed that the barrier for a direct inter-subunit proton transfer is possibly too high for supporting a complete proton transfer between the subunits[31], whereas the complete proton transport between the ion-pair elements and central proton pathway is further hampered by bulky residues (e.g. Trp213 of Nqo13, Trp225/Trp131 of Nqo12). Recent QM/MM calculations[55], suggested that the cryoEM structure of *Yarrowia* Complex I could support horizontal proton transfer between the Nqo11-14 and Nqo14-13 subunits interface, if the ion-pair lysine is modelled in a neutral state (Lys⁰). Although such (Glu⁻/Lys⁰) configurations are electrostatically unlikely, we note that the barrier for the proton transfer along the Nqo13-Nqo12 (ND4-ND5) interface and further across the membrane remains unknown, hampering systematic testing of the ND5-only pumping model[19,20].

Interestingly, despite the well-wired P-side connections in Nqo12 and the more non-polar P-side exit pathway in Nqo13, we observed no significant overall differences in the ΔpH-mediated proton conduction rates between the constructs, showing that both subunits support proton transport across the membrane. However, as Nqo12 may initiate the proton release across the membrane (Fig. 5b)[1], it could benefit from kinetically more efficient proton release relative to the other putative proton release sites at the Nqo12-13, Nqo13-14, and Nqo14-10/11 interfaces (Fig. 5b). In this regard, we found that E132Q[12] is indeed kinetically highly efficient and nearly outcompetes the proton pumping efficiency of ATP synthase under the studied conditions. The terminal Asp386 of Nqo12 is in an analogous position to Phe378 of Nqo13 (Supplementary Fig. 1i) and could act as a local gate for controlling the proton release by transient wetting transitions. At a general level, these findings are in line with proton conduction principles found in artificial proton channels, where the transport also takes place by transient water wires, which provide elements for conducting the

protons across dry patches of the proteins[56]. Despite significant differences in the protein architecture, these designed systems show an interesting resemblance to protonation dynamics in the antiporter modules, which also employ transient water wires to transport the protons across the membrane.

The proton pumping in Complex I could be initiated by conformational changes in charged networks triggered by the movement of the quinol species from the hydrophilic domain to a second membrane-bound Q binding site[1,9,22,33] that in turn triggers lateral proton transfer reactions towards the Nqo11/Nqo14 interface[35,36] (Fig. 5b). This protonation cascade leads to stepwise opening of ion-pairs that favour the lateral proton transport towards the next antiporter, and leading to the propagation of protonation signal via Nqo14 and Nqo13 to Nqo12. The well-wired P-side connections in Nqo12[19,20,28,30,34] could initiate proton release across the membrane (see above). The proton release increases the proton affinity of the "middle Lys", favouring closing of the ion-pair at the Nqo12/Nqo13 interface, and releases the "Nqo13 proton" across the membrane. Re-protonation of Lys235 and closing of the Glu123-Lys204 ion-pair similarly eject the "Nqo14 proton" across the membrane, followed by propagation of the signal to the membrane-bound Q-binding site. This re-sets the machinery by releasing the quinol and leading to uptake of a new quinone from the membrane pool. The proton transport in the antiporter modules supports that the ion-pairs modulate the proton transfer barriers, as well as central residues along the putative proton pathways. Taken together, we suggest that the isolated antiporters also utilise similar conformational changes in the ion-pairs and wetting/drying transition to channel the protons across the membrane (Fig. 5a).

The coupling between the individual charged elements within and between neighbouring subunits is essential for gating the proton transfer reactions. If the coupling is disturbed, this could lead to a proton leak and dissipation of the energy, instead of a pumping step, as suggested by inhibition of the coupled proton pumping and oxidoreductase activity[10–17]. In terms of our wave propagation model, the

perturbed coupling between the elements could hamper the backward wave propagation to the membrane-bound Q site, and an inhibition of the proton pumping machinery. We speculate that severe mutations completely block the wave propagation in both forward and backward directions at the site of the mutation (leading to a pumping stoichiometry of 0[17]), whilst softer mutations could allow for the wave propagation without loss of energy, at least with modest PMF conditions (and lead to a pumping stoichiometry of 4[17]).

During assembly of the Complex I machinery, the individual antiporter modules are stepwise put together to secure correct subunit interactions. However, it remains important that the individual antiporter modules do not dissipate the *pmf* that could hamper the energy metabolism. Indeed, the *E. coli* growth is severely hampered in minimal media (Supplementary Fig. 10m, n), suggesting that the antiporter modules might dissipate part of the *pmf* during the growth, although it is difficult to draw detailed conclusions of the dissipation effects in vivo conditions based on these observations. We note that the Nqo12-Nqo13 contacts are established rather late during the assembly pathway of Complex I[57,58], and couples to the release of several assembly factors[57]. In the light of current results, we suggest that these assembly proteins could modulate the rate of proton conduction of the emerging Complex I, and prevent the dissipation of the *pmf* by blocking the native antiporter contacts, until all components gating the redox-driven proton pumping process are in place.

In addition to their key role in the antiporter modules of Complex I, buried ion-pairs establish functional motifs also in several other enzymes. For example, the redox-triggered opening of the Asp61-Lys317 ion-pair next to the oxygen evolving $Mn_4O_5Ca$ cluster of Photosystem II was recently suggested to create electric fields that direct the stepwise deprotonation of substrate water molecules[59], *cf.* also[60,61], whilst the D-propionate/Arg438 ion-pair could serve a similar function in cytochrome *c* oxidase[62]. Ion-pair dissociation was also found to regulate ATP hydrolysis in the molecular chaperone, Hsp90[63], whereas conformational changes in the Arg159-Glu58 ion-pair in ATP synthase drive the protonation and rotatory motion of the c-ring[64]. Understanding the physical properties of buried ion-pairs has also stimulated the design of buried networks of ion-pairs in artificial proteins[65] as well as in natural proteins[66], highlighting the delicate balance between electrostatic and solvation effects that must be carefully tuned to enable proton transport and conformational switching in Complex I.

In summary, we have shown here using a multi-layered experimental and computational approach that the isolated antiporter modules of Complex I contain all necessary elements to transport protons across proteoliposome membranes. We further found that the rate of proton conduction is controlled by the conformational state of a buried ion-pair, with electric field effects and wetting transitions forming a basis for the gating mechanism. Mechanistic studies of the isolated antiporter modules could provide central understanding of how key residues involved in, *e.g.*, mitochondrial diseases affect the elementary proton transport properties, which are difficult to assess in the native intact Complex I. Taken together, our findings show how the antiporter modules and their interactions gate the proton pumping in Complex I, highlighting key functional principles of its unresolved long-range energy transduction mechanism.

## Methods

### Molecular dynamics simulations

Classical MD simulations of the antiporter-like constructs were performed based on coordinates of the Nqo13 and Nqo12 subunits extracted from the x-ray structure of Complex I from *T. thermophilus* (PDB ID: 4HEA[23]), followed by removal of the long transverse helix in Nqo12 (residues 516-606). In silico mutations were generated using

PyMOL[67] by selecting the lowest energy rotamers. The protein constructs were embedded in either a POPC or in a POPE/POPG/cardiolipin (67%, 23%, and 9.8%) membrane together with water molecules and with 150 mM NaCl concentration. The final systems comprised of *ca.* 59,400-160,000 atoms. The systems were simulated for 1000 ns in duplicates at $T = 310$ K using a 2 fs timestep with the CHARMM36m force field[68], and treating the long-range electrostatics by the Particle Mesh Ewald approach. The MD simulations were performed using NAMD v.2.14/v3.0[69]. Visual Molecular Dynamics (VMD)[70], UCSF Chimera[71], Chimera-X[72] and MDAnalysis[73] were used for visualisation and analysis. p$K_a$ shifts were estimated based on electrostatic calculations in combination with Monte Carlo sampling (see Supplementary Information, Supplementary Methods), with results shown in Supplementary Fig. 16 and Supplementary Table 3. See also Supplementary Fig. 1 for system setup, and Supplementary Table 4 for details of the MD simulations.

### Hybrid QM/MM calculations

QM/MM calculations were performed to explore the proton transfer energetics in the Nqo13 and Nqo12$^{\Delta TH}$ constructs. To this end, classically relaxed structures showing water connectivity across the lateral proton transfer pathway, and from the N-side to the middle Lys, were selected. To explore the lateral proton transfer reactions, we selected a QM region comprising *ca.* 130 atoms in Nqo13 and *ca.* 150 in Nqo12 (see Supplementary Table 9). The QM/MM boundary was described by link atoms, introduced between the C$\beta$ and C$\alpha$ atoms. The reaction coordinate for the lateral proton transfer process, $R$, was described as a linear combination of the breaking bonds and forming bonds (Supplementary Fig. 6). A minimum energy reaction pathway (MEP) was optimised along $R$, by applying harmonic restraints using a force constant of 3000 kcal mol$^{-1}$Å$^{-2}$. Based on the MEP, free energy profiles for the lateral proton transfer reaction were computed using umbrella sampling/weighted histogram analysis method (US/WHAM)[74,75] by restraining each window to their corresponding $R$ value with a harmonic force constant of 100 kcal mol$^{-1}$Å$^{-2}$, at $T = 310$ K. Atoms around 10 Å of the QM region, were allowed to move, while keeping the remaining MM system fixed. Free energy profiles were computed based on 173 ps sampling for the Nqo13 constructs (75 windows x 2.3 ps/window), and 120 ps for the Nqo12$^{\Delta TH}$ constructs (52 windows x 2.3 ps/window). QM atoms were modelled at the B3LYP-D3/def-SVP level[76–79], and MM atoms at the CHARMM36 level[68]. Electric field effects along the proton wire were estimated in the QM/MM models for WT-Nqo13 and Nqo12$^{\Delta TH}$, using the same QM-MM partition as in the free energy calculations, by varying the ion-pair distance (Glu123-Lys204 in Nqo13, and Glu132-Lys216 in Nqo12). The energetics were benchmarked against the correlated RPA level[80] (see Supplementary Fig. 9 and Supplementary Table 8 for details), whilst entropic effects were accounted for by the classical MD sampling. US and electric-field QM/MM calculations were performed by coupling CHARMMc38b1 and TURBOMOLE v.7.3–7.5[81–83] together.

To estimate the free energy landscape of the proton uptake from N-side (Lys282 to Lys235 in Nqo13), we used a QM/MM simulation together with the recent shared-bias well-tempered metadynamics-extended adaptive biasing force (MWE) method[84,85]. Here, the reaction phase space was explored by a harmonic potential $B(\xi(x),\lambda)$, coupled to a moving fictitious particle $\lambda$, with an additional time-dependent bias potential $B_{MWE}(\lambda,t)$ used to ensure a uniform sampling. Parallel sampling was achieved by performing the simulation with 10 walkers (bin Width=0.1 Å), each sampled for 22.5 ps with a 0.5 fs timestep and at $T = 310$ K. Gaussian hills with heights and widths of 1.0 kJ mol$^{-1}$ and 0.2 Å, respectively, were added every 20 fs. The QM region comprised *ca.* 150 atoms (see Supplementary Table 9). The reaction coordinate (CV) was defined using the modified centre of excess charge (mCEC), with the sidechain nitrogen atom of the residues Lys282 (N-side) and Lys235 (middle Lys) set to donor and acceptor, respectively, spanning the CV

range [0,10 Å]. The QM/MM-MWE-sampling was performed using OpenMM/Fermions + +[86] with the unbiased PMF retrieved by MBAR[87] analysis.

## Cloning, expression, mutagenesis, and purification

Nqo12 and Nqo13 subunits from *Thermus thermophilus* HB27c Complex I were isolated from the genomic DNA and inserted in the pWALDO expression vector[88] by Gibson assembly. The expression vector contained an ampicillin gene resistance, a TEV cleavage site and a sequence coding for the superfolder GFP (sfGFP). A GSAGS linker was inserted between the C-terminus of Nqo12 and the TEV cleavage site, with residues 516–605 (transverse helix) removed for Nqo12$^{\Delta TH}$. Due to the different topology of Nqo12 and Nqo13, the cleavable sfGFP is located on the P-side of Nqo13 and the N-side of the Nqo12$^{\Delta TH}$. Single point mutations were created on both Nqo12 and Nqo13 using designed primers (see Supplementary Table 5, *ThermoFischer*), with mutations confirmed by DNA sequencing (Eurofins, Uppsala, Sweden).

The different Nqo-pWALDO constructs were transformed into *E. coli* Lemo21[89] competent cells for over expression. A pre-culture was inoculated into TB medium supplemented with 0.1mM L-rhamnose and 100 μg mL$^{-1}$ ampicillin, as well as 17 μg mL$^{-1}$ chloramphenicol at 37 °C until OD$^{600}$ = 0.5 before inducing with 0.4 mM IPTG overnight at 25 °C. The cells were collected by centrifugation for 15 min at 6000 × *g*, and lysed using an Emulsiflex. Unbroken cells were removed by centrifugation for 20 min at 25,000 g, and membranes were collected by centrifugation at 180,000 × *g* for 2 h. Membranes were resuspended in 25 mM HEPES pH=7.5, and 150 mM NaCl at 6 mg mL$^{-1}$. Membrane proteins were solubilised using 0.8% DDM for 1 h at 4 °C, and collected by ultra-centrifugation for 30 min at 180,000 g. The supernatant was then loaded in 5 mL of Ni-NTA resin (ThermoFischer), to bind for one hour under agitation. Prior to this, the resin was equilibrated in 25 mM HEPES pH=7.5, 300 mM NaCl, 0.05% DDM. After binding, the sample was let through the column, and washed with 10 CV of the equilibration buffer supplemented with 20 mM imidazole. Multiple wash steps were performed using 75 mM, 100 mM, and 150 mM imidazole to wash off contaminants, before eluting using the equilibration buffer supplemented with 300 mM imidazole. The elution fraction was concentrated down to 1 mL and loaded in a HiLoad 16/600 Superdex 200 pg column (Cytiva) and eluted using 25 mM HEPES pH=7.5, 150 mM NaCl, 0.05% DDM.

## Growth tests of the antiporter subunits

The expression conditions of WT Nqo12 and Nqo13 were tested as described above both in TB medium and M9 minimum medium. The results of the growth test are shown in Supplementary Fig. 10m, n.

## Expression and purification of ATP synthase

*E. coli* ATP synthase was purified using a protocol adapted from ref. 48. To this end, the pFV2 plasmid was transformed into *E. coli* DK8 competent cells. The cells were grown by first inoculating a single isolated colony in 3 mL of LB, supplemented with ampicillin (100 μg mL$^{-1}$). After 3 h shaking at 37 °C, the inoculation is transferred into 60 mL of fresh LB for 4 h before the transfer into 6 L of LB, supplemented with 100 μg mL$^{-1}$ ampicillin and 1 mM MgCl$_2$. Flasks were left shaking overnight at 37 °C. Cells were collected by centrifugation for 20 min at 11,000 × *g* and resuspended in lysis buffer (50 mM HEPES pH 8.0, 100 mM NaCl, 5% glycerol), before lysing using an Emulsiflex. Unbroken cells were removed by centrifugation for 30 min at 12,000 × *g*, 4 °C, and membranes were collected by centrifugation for 1 h at 250,000 × *g*, 4 °C. Membranes were resuspended (1.5 mL g$^{-1}$ of cells) with extraction buffer (50 mM HEPES pH 7.5, 200 mM KCl, 150 mM sucrose, 0.8% Type II-S soybean lipids, 1.5% octyl-β-glucoside) and solubilised for 1 h under mild agitation at 4 °C. Insolubilised

membranes were removed by centrifugation for 30 min at 250,000 × *g*, 4 °C. The supernatant was then loaded twice at 1.5 mL min$^{-1}$ in a 5 mL His-Trap HP column (Cytiva), that was previously equilibrated with 3 CV of purification buffer (50 mM HEPES pH 7.5, 200 mM KCl, 150 mM sucrose, 0.005% lauryl maltose neopentyl glycol (LMNG), 20 mM imidazole). The sample was washed using the purification buffer with increasing steps of imidazole concentrations (20 mM, 40 mM, 90 mM). Elution was performed using the purification buffer supplemented with 250 mM imidazole. Relevant fractions were pooled and concentrated using a concentrator with a 100 kDa cut-off. The concentrated sample was loaded in a HiLoad 16/600 Superose 6 pg (Cytiva) for size-exclusion chromatography using SEC buffer (50 mM HEPES pH 7.5, 200 mM KCl, 150 mM sucrose, 0,005% LMNG). Relevant fractions were pooled and concentrated using an Amicon Ultra Protein Concentrator 100 K (MWCO). The protein concentration was measured using a NanoDrop spectrophotometer.

## Proteoliposome preparation

*E. coli* polar lipids (Avanti Polar Lipids) were dried under nitrogen steam until forming a thin layer of lipids, followed by desiccation under vacuum overnight. The dried lipids were resuspended at 5 mg mL$^{-1}$ using the proteoliposome buffer (2 mM MOPS/KOH pH 7.2, 50 mM KCl). For preparation of unilamellar vesicles, the multi-lamellar vesicles were tip-sonicated for one minute. The liposomes were extruded 21 times through a 0.1 μm membrane (Nuclepore membranes, Whatman Ltd.). The proteins were reconstituted using 0.4% final concentration of cholate, prior to addition of the protein sample to the lipids/detergent mixture, which were incubated for 30 min on ice, to incorporate one Nqo protein per vesicle. Cholate was removed with a prepacked PD-10 column (Cytiva), and the proteoliposomes were eluted with 1.5 mL of proteoliposome buffer. A proteoliposome diameter size of 60 nm was used for the acid-induced proton conduction assays, and 200 nm for the ATPase-driven assays (see below), as determined by dynamic light scattering analysis.

Type II-S soybean lipids were used for preparation of ATP synthase-liposome due to their compatibility with ATP synthase[90]. To this end, type II-S soybean lipids (ThermoFischer) were dissolved at 5 mg mL$^{-1}$ into the ATP synthase-liposome buffer (10 mM Hepes/KOH pH 7.5, 100 mM KCl, 5 mM MgCl$_2$). Dissolved liposomes were frozen and thawn 7 times to break multi-lamellar vesicles, and then extruded 21 times through a 0.2 μm membrane (Nuclepore membranes, Whatman Ltd.). Reconstitution was performed in 550 μL final volume including 400 μL of extruded lipids, 0.6% cholate, ATP synthase and Nqo subunits or AqpZ in a 1–5 molar ratio. PD-10 (Cytiva) was used to remove cholate. Proteoliposome were pelleted by centrifugation for 30 min, 4 °C, 150,000 g and resuspended in 100 μL of liposomes buffer.

## Antiporter orientation in proteoliposomes

The orientation of the Nqo modules in liposomes was determined by NTA-Atto 647 N labelling (Sigma-Aldrich). 180 μL of proteoliposomes were incubated with an equal volume of DDM or buffer for an hour prior to the addition of a 1:150 molar ratio protein to NTA-Atto label for 30 min to let the NTA group interact with the hexa-His-tag. Excess dye was removed by PD SpinTrap G-25 desalting column (Cytiva). The fluorescence emission scan was measured upon excitation at 647 nm. The orientation ratio was determined based on *n* = 3 independent measurements at 662 nm.

## Proton conduction assays

The proton conduction assays were developed based on a previously used protocol for quantification of protonation dynamics in *bo*$_3$ oxidase[91]. To this end, 5 mM of the pH-sensitive dye molecule pyranine (8-hydroxypyrene-1,3,6-trisulfonic acid, HPTS, Sigma-

Aldrich) was incorporated inside the proteoliposomes by three freeze-thaw cycles, followed by removal of non-incorporated HPTS with a PD-10 column (Cytiva). The liposomes were eluted using 1.5 mL of proteoliposome buffer. 30 μL of PLs were diluted up to 1 mL with proteoliposome buffer, initiating the dissipation of the initial ΔpH across the PL bilayer, and 4 nM valinomycin that affect the initial baseline, whereas the final steady-state pH (reported in Fig. 3d, e), obtained prior to addition of nigericin, is independent of the baseline. The proton conduction was assayed at 37 °C by measuring the change in the HPTS fluorescence emission at 510 nm upon excitation at 404 nm and 454 nm using a fluorescence spectrophotometer (Cary Eclipse Fluorescence Spectrophotometer, Agilent Technologies) and plastic cuvettes ($d = 1$ cm). The reaction was started by addition of potassium acetate or methylamine hydrochloride to create a ΔpH gradient across the liposome membrane. The proton gradient was disrupted by the addition of 2 μL 2 μM nigericin (solved in absolute EtOH). ΔpH and ΔΨ-mediated ion-transport was performed as described in ref. 48. To this end, the ΔpH-mediated proton conduction was tested using 30 μL of proteoliposomes formed in 2 mM MOPS-NaOH pH 7.2, 2.5 mM MgCl$_2$, 50 mM KCl, and diluted to 1 mL with 2 mM Bicine-NaOH pH 8.4, 2.5 mM MgCl$_2$, 50 mM KCl. The ΔΨ-mediated proton conduction was tested by using 30 μL of proteoliposomes formed in 2 mM MOPS-NaOH pH 7.2, 2.5 mM MgCl$_2$, 50 mM NaCl, 1 mM KCl, and diluted to 1 mL in 2 mM MOPS-NaOH pH 7.2, 2.5 mM MgCl$_2$, 34 mM NaCl, 16 mM KCl. 4 nM valinomycin was added after 1 min to initiate to the proton conduction (see Supplementary Fig. 13b, c). The fluorescence signal was converted into a pH change, based pH calibration (Supplementary Fig. 11a) using the following relation,

$$pH = a \log_{10}(I_{404nm}/I_{454nm}) + b \qquad (1)$$

where $a$=1.1684 and $b$=7.721.

The proton conduction for ATP synthase was also assessed by ACMA fluorescence. In this regard, 20 μL of proteoliposomes, 4 nM valinomycin, and 4 μM ACMA were added to a final volume of 1 mL. The reactions were started by addition of 0.2 mM ATP, followed by dissipation of the ΔpH by nigericin. Both GFP-cleaved (with TEV protease) and uncleaved constructs resulted in the same proton conduction rates (Supplementary Fig. 14c, d, g).

The proton conduction was assayed at 37 °C by measuring the change in the ACMA fluorescence emission at 480 nm upon excitation at 410 nm using a fluorescence spectrophotometer (Cary Eclipse Fluorescence Spectrophotometer, Agilent Technologies) and plastic cuvettes ($d = 1$ cm). The relative fluorescence was plotted as an average of the individual measurements, with the baseline (Δ$F$ = 1.0) level obtained by averaging over the first minute of the ACMA fluorescence signal. Initial rates were obtained by fitting the data with a linear decay ($a_0 + k_1 t$) in python using matplotlib, scipy, and numpy (Supplementary Fig. 15). The data were measured in $n$ = 3–6 independent measurements. The time-resolution of the ACMA fluorescence experiments is 100 ms, and 600 ms for the pyranine experiments.

### Reporting summary
Further information on research design is available in the Nature Portfolio Reporting Summary linked to this article.

## Data availability
Source data are provided as a Source Data file. Additional data supporting the findings of this manuscript are available from the corresponding authors upon request. The PDB code of the previously published structure used in this study is 4HEA. The multiscale simulation data generated in this study have been deposited in the Zenodo database under accession code https://doi.org/10.5281/zenodo.13732353. Source data are provided with this paper.

## Code availability
The codes used for the simulations and analysis are commercially (TURBOMOLE[81,82], CHARMMc38b1[83], Fermions + +[86]) or freely (NAMD[69], VMD[70], UCSF Chimera[71], Chimera-X[72], MDAnalysis[73]) available.

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

## Acknowledgements

We thank Prof. Christoph von Ballmoos for kindly providing the pFV2 plasmid, and Prof. James Sturgis for kindly providing the AqpZ plasmid. This work was supported by the European Research Council (ERC) under the European Union's Horizon 2020 research and innovation programme (grant 715311), the Swedish Research Council (VR), and the Knut and Alice Wallenberg (KAW) Foundation (grant 2019.0251). Computing resources were provided by the National Academic Infrastructure for Supercomputing in Sweden (NAISS 2023/6-128) and the Swedish National Infrastructure for Computing (SNIC 2022/1-29, 2021/1-40, SNIC 2021/1-60, 2022/13-14), and the Leibniz-Rechenzentrum (LRZ, project: pr83ro).

## Author contributions

V.R.I.K. designed the study; A.B., S.B. constructed, isolated, and experimentally characterised the Nqo models; A.B., J.Z., J.B. G.K. developed molecular biology for construct design; A.B., S.B., J.B., G.K., V.R.I.K developed proteoliposome assays; S.B. and V.R.I.K. developed ATPase assays, A.B., S.B. collected proton conduction data; P.S., H.K., D.A., performed MD and QM/MM simulations; A.P.G.H. performed electrostatic calculations and cphmd simulations; A.B., P.S., S.B., H.K., J.Z., D.A., A.P.G.H., J.B., G.K., V.R.I.K. analysed the data; V.R.I.K. directed the project; V.R.I.K. wrote the manuscript with input from all authors.

## Funding

## Competing interests

The authors declare no competing interests.
