## [Peer Review File · Nature Communications]

Dissected Antiporter Modules Establish Minimal Proton-Conduction Elements in the Respiratory Complex IREVIEWER COMMENTS

Reviewer #1 (Remarks to the Author):

The work entitled “ Dissected Antiporter Modules Establish Minimal Proton-Conduction Elements in the Respiratory Complex I” presents a combined computational, using MD and QM/MM, and experimental study of a minimal proton-conducting membrane modules, created by the authors by engineering and dissecting the key elements of the bacterial Complex I, the highly intricate redox-driven proton pump that powers oxidative phosphorylation across all domains of life. In the work, the authors show that the isolated antiporter-like modules are able to conduct protons across proteoliposome membranes, and present evidence that the rate of proton conduction is controlled by conformational changes of buried ion-pairs that modulate the reaction barriers by electric field effects.

Overall the work is well done, figures are impressive and the researchers have a long tradition and experience in working with complex I, which is a relevant topic. The results are reasonably well supported by the presented data. In summary the relevance and quality of the work supports its publication. However, the work lacks in clarity and novelty, and therefore my overall feeling is that it could be better suited for a more specialized journal. Specific comments are described below.

The two main issues with the present work are 1) novelty, and 2) clarity for the general readership.

1) The work main aim is to provide evidence supporting that the proton pumping is catalyzed by the antiporter-like subunits Nqo12, Nqo13, and Nqo14, together with Nqo7, through an S-shaped water mediate proton wire across the membrane, which, already in previous works were modulated by the conformational state of the conserved Glu-Lys ion pairs located at the interface of each antiporter-like subunit. Moreover, the proposal that the long-range proton transport process takes place by an electric wave that propagates in forward and reverse directions across the 200 Å wide membrane

domain, similar to an 'electrical cradle' was also proposed in previous works.

Since the role of the S shaped tunnel, the ion pair and the electric wave were already proposed, and supported in previous studies it is not clear, beyond providing support for them what is the general novelty of the present work.

2) The work is very hard to follow for general readers which are not specialists in Complex I structural biochemistry. Several key aspects are not explained, and should be. For example, i) how did the authors come to the hypothesis of the key ion pair as being the controller of the process. ii) The hypothesis of the propagating electric wave should be explained.

Concerning the methodology, I will focus on computer simulations

3) The authors computed using QM/MM a minimum energy reaction pathway (MEP) followed by umbrella sampling the proton transfer reaction. For this reaction, the reaction coordinate is never presented nor explained. More important, if I understood correctly only one step of proton transfer

was studied, and not the whole path, or at least all key proton transfer steps. In this sense the work seems to be incomplete.

4) Concerning the classical MD simulations, and the observed hydration of the S shaped proton channel. What I feel is missing is an evaluation of the pKa of key residues along the channel, how the change in their protonation state results in small conformational changes (i.e. side chain movements), and how this changes due to the electric wave propagation model, or more general the effect of ET from NADH to Q.

5) The coupling with the electric field, modified and generated by the ion pair and ultimately by the electron movement from NADH to Quinone is not clear at all. This is central to the work and should be clearly explained.

Reviewer #2 (Remarks to the Author):

The paper explores the properties of two membrane subunits, Nqo 12 and 13, in respiratory Complex I, responsible for transmembrane proton transfer. From the finding that these isolated and reconstituted subunits increase the membrane proton permeability, the authors suggest that these isolated and reconstituted subunits comprise all necessary functional elements for conducting protons across membranes. However, the conclusion and experimental approach appear to lack logical coherence.

Given that the studied subunits are exposed to high electrochemical proton potential in situ (up to 200 mV in mitochondria), they should not induce proton leakage, as this would establish a futile cycle. Proton channels in Nqo12 and 13 should open only in response to the wave of electrostatic interactions initiated by quinone reduction. The authors acknowledge this in the discussion, and to overcome this problem they propose that in vivo assembly proteins prevent proton dissipation by blocking native antiporter contacts. Isolated and reconstituted without assembly proteins, Nqo12 and 13 might produce proton leaks not through their channels but due to the loss of correct surrounding or folding.

Wild-type Nqo12 and 13 exhibit proton permeability slightly higher than the natural permeability of the liposome membrane. Wild-type Nqo12 and 13 exhibit proton permeability slightly greater than the natural permeability of the liposomal membrane. Mutations leading to conformational changes lead to a significant increase in proton fluxes; according to the authors, this indicates the control of proton channel by replaced amino acid residues. However, there is no evidence whether proton flux is mediated by a natural proton translocation mechanism or is due to destabilization of the lipid environment by a misfolded protein creating an artificial proton pathway.

Regarding the results, there are several comments on the treatment of biochemical data. The initial phase in Fig 3b is not resolved. A major part of this phase is formed by the response of not inner, but outer pyranine since the removal of the outer dye never can be 100%. This means that the initial phase of the proton flux is lost and 5-digit numbers of k_2 (Supplementary Table 1) do not have any meaning.

The titration curve of pyranine with a pKa over 8 (Supplementary Fig11a), inconsistent with the

known pKa of 7.2, raises questions.

The fitting of curves is compromised by poor time resolution, with initial experimental points often deviating from fitted curves. However, some experimental results, such as the addition of NaOH, methylamine, and acetate, seem excessive and do not provide new information beyond what is demonstrated with co-reconstitution with the slow-operating ATPase. These experiments show quite clearly that Nqo12 and 13 cause some proton leakage; the more these proteins are disrupted by mutations, the greater the leakage of protons.

In the introduction, the mention of long-range proton transfer should reference the concept of the electrostatic interactions wave in Complex I, first suggested in 2008 by Euro et al. (BBA 1777, p1166) and described in more detail in 2013 by Verkhovskaya and Bloch (Int L Biochem Cell Biol. 45(2): 491).

In summary, the paper is overly verbose, containing excessive data and an overload of figures and numbers in the biochemical part. Streamlining the content and focusing on key findings could enhance clarity and impact.

Reviewer #3 (Remarks to the Author):

Report

Begiah et al describe their ambitious efforts to deconstruct a bacterial complex I (Thermus thermophilus) into parts and check for their remaining function. The focus is on the two membrane embedded antiporter like modules nqo12 and nqo13, which they successfully express in E. coli and purify via affinity chromatography. To facilitate expression, they remove the transversal helix from nqo12 and they can convincingly show that they have expressed and purified the two proteins. To get an idea how these proteins could behave isolated in membranes, they perform molecular dynamics simulation studies of the individual subunits in lipid bilayers, either in pure POPC or the more natural variant POPE/POPG/CDL. While they show the simulation data, they unfortunately do not discuss if the observed differences are relevant, or if the choice of lipids does not make a big difference. For their liposome assays, they settle for E. coli polar lipid extract, which is closer to second setting (see more on this below).

They identify several residues that are expected to influence the activity of the proton path, namely the intersubunit bridging residues (via salt bridge) or polar residues along the proton paths. No further explanation is given on which basis the replaced amino acids were chosen. While an exchange of a glutamate with a glutamine (as in E123Q or E132Q) is not surprising, an exchange of a lysine with an isoleucine (as in K385I) deserves more attention as it might also affect the overall protein stability and functionality.

In their main experimental part, they perform three types of proton conducting measurements across the membrane.

1. The establish a ΔpH by adding base NaOH to the outside of the liposome, thereby creating an driving force for outwards proton transport, which they follow using the pH sensitive dye pyranine entrapped in the liposomes. In their liposomes they reconstitute either different Nqo variants, or AqpZ and no protein as negative controls. Throughout their experiments, no difference between

empty liposomes or liposomes containing AqpZ are observed.

While they see differences between the negative controls and their Nqo variants, the majority of the signal seems unspecific, making interpretation difficult. They observed a biphasic behaviour of the transport, however they do not comment on the relevance or origin of the slower transport rate. The display of the data is somewhat unfortunate, as the kinetics of the rapid (but more relevant and later discussed) phase is not resolved. The author might consider using a different form, e.g. broken x-axis or two graphs with different time scales.

2. Next, they switch to an approach, in which they add potassium acetate to the liposome. The expectation is that acetate will be protonated to a small degree ($pK_a \sim 4.8$) at pH 7.2, and in its protonated form it will diffuse across the liposomal membrane. Once inside the liposome, the same dissociation distribution applies and most of the transported acetic acid will be present in its ionized form, releasing the proton and thus acidifying the liposomal lumen. They indeed see a rapid decrease in the lumen of the liposomes, and they find that this drop is less pronounced in liposomes containing Nqo compared to control liposomes (empty or AqpZ). The rationale is that Nqo allows for rapid backflow of protons and thus a decreased ΔpH is established.

In their supplementary figures, they show a similar experiment, in which they use methylamine ($pK_a \sim 10.6$) to establish an inverse pH gradient (inside basic) and again observe competitive Nqo proton transport. Again, the display of the data does not allow to resolve the kinetics and the conclusions are solely drawn on the amplitude of the total drop (i.e. the resulting ΔpH within the different variants). The difference between empty liposome and liposomes containing Nqo variants are small and hard to appreciate, but they behave in the expected manner.

3. In a third approach, they use purified E. coli ATP synthase and co-reconstitute with the different variants. Instead of pyranine as used above in the two previous approaches, they use ACMA which is traditionally applied to monitor acidification of the luminal pH in liposome experiments, which is observed as a quench in fluorescence. They find less quench in liposomes containing Nqo upon addition of ATP compared to empty liposomes. The rationale is that the co-reconstituted Nqo prohibits the build-up of a large ΔpH by parallel extrusion of the ATP synthase pumped protons via the co-reconstituted Nqo. In contrast to the previous measurements, no “background signal is observed in the empty liposomes”, giving clearer distinction between liposomes having proteins reconstituted. Unfortunately, only normalized data are shown and the actual extent of ACMA quench is not visible. The relative fluorescence goes to 0 (or even below in 4f, blue trace) which makes not much sense in the case in an ACMA experiment (in the reviewer’s experience, even a perfect ACMA quench experiments leaves 5 to 10% fluorescence behind). I don’t fully understand what is set as the lower value of the normalized fluorescence (The minimal value of wt trace????). That should be explained better and/or the data should be shown unprocessed.

This is a very impressive body of experimental work and it is neatly documented, starting from expression, purification and proton transport measurements in liposomes.

All three types of measurements described above show the same principal outcome.

- The wt works as proton translocating unit in all three assays
- The glutamic acid residue at the intersubunit bridge seems to control the path, and replacement with glutamine seems to keep the pathway more in the “open” conformation, and the proton transport is accelerated.
- Replacement of a lysine at the exit pathway by an isoleucine seems to decrease the rate of proton

transport, potentially by decreasing hydration of the local environment or steric hindrance. Taken together, I tend to trust in the presented data, that the data show ΔpH driven transport of the individual subunits and that the mutations made affect the transport activity in the proposed fashion. However, it is unsatisfying that the kinetics of the proton transport is not better resolved. Although the determined rates (the unit is not quite clear to me, but I suspect pH units per min), the interpretation is based on the build-up of different steady state levels after a rapid proton transport. The role of second slow kinetic phase, which is well resolved, is not further discussed (but is likely also not relevant).

If the findings are correct, the present manuscript makes an important contribution to solve the current problem of proton transport pathways in complex I, and its data argue strongly against the ND5-only theory by showing that the individual units are able of proton transport. Currently, functional measurements cannot keep up with the speed of new high-resolution structures, and it is important to keep these at a balance. The present study using state-of-the-art MD simulations to make predictions and test them in a well-controlled minimal setup is a brave and powerful attempt to understand the molecular mechanism of proteins like complex. In a very simplified form, an earlier study by Gemperle et al (<https://pubmed.ncbi.nlm.nih.gov/17583799/>) also investigated transport activity of complex I subunits. There, the same subunits (NuoL/N) were expressed in yeast and membrane fraction enriched with the protein were probed for $\text{H}^+/\text{Na}^+/\text{K}^+$ antiport. The present study however goes far beyond that earlier work.

The paper is well written, the introduction is informative and after a very dense simulation section, the experimental work is well described (see below for a few comments).

The discussion is kept straightforward by arguing that the present data argue against the ND-5 model. They argue that both subunits show similar transport rates, supporting the idea that they might work as a tandem. A interesting question, why mutations in the native complex I favors either 0 or 4 pumped protons (as described in ref. 16) was not touched in the discussion. It would have been interesting how the author relate to the connection between two subunits.

Towards the end, the discussion is lead towards the direction of the assembly of the complex and the observed transport rates and I was stumbling on the following sentence:

“In the light of current results, we suggest that these assembly proteins could modulate the rate of proton conduction of the emerging Complex I, and prevent the dissipation of the pmf by blocking the native antiporter contacts, until all components gating the redox-driven proton pumping process are in place.” (Beghiah et al., p. 12).”

If isolated and correctly reconstituted subunits nqo12 or 13 were indeed capable of dissipating the pmf, why did it then not happen during the overexpression culture, where these proteins were overexpressed into the cytoplasmic membrane? Maybe a growth test in more challenging media than LB would show such a phenotype?

The mode of dissipation would be $\Delta\psi$ driven proton transport, which might waste the essential potential otherwise used to drive important processes such as nutrient uptake or waste export by secondary active transporters. However, this would imply that the typical mode of action would be $\Delta\psi$ driven transport and not ΔpH driven transport, as the ΔpH is negligible and can even be inverted (e.g. growing *E. coli* at pH 8).

It seems thus crucial to me that the authors connivingly show that the purified subunits are capable to drive pure potential driven proton transport. This has been shown for isolate Fo of ATP synthase (ref. 49) or Na^+/H^+ antiporters. The details of such experiments are described below.

Comments (some of it has been written before the text above and might contain repetitions):

In the following, I list some of my more technical comments on the experiments and suggestions for experiments that might help to alleviate some of the limitations. In my opinion, certain experiments need to be done before the manuscript is ready for publication.

1. The authors have used valinomycin in all experiments, with identical potassium concentration on either side. The rationale is to relieve a potentially opposing membrane potential that would block proton transport. As a consequence, the observed ΔpH tend to become larger, rates are increased and signals become easier to detect. While this is a correct observation and application, it would be very interesting to see, if the presence of valinomycin and potassium makes an actual difference.

- This would allow to draw direct conclusion if Nqo catalyze electrogenic or electroneutral proton transport. In the native complex I, the transport is electrogenic, and thus omission of valinomycin makes a difference. If however, no difference is observed, isolated Nqo are either distorted and catalyze electroneutral proton transport (e.g. via leaky Cl^- cotransport) or the liposomes are leaky and a stable $\Delta\psi$ cannot be held (see below).

2. In all the experiment, a ΔpH is used a driving force. This is somewhat counterintuitive, as the main driving force in e.g. reverse electron transfer is the membrane potential. I therefore strongly suggest driving proton transport by a membrane potential only that can be produced using a potassium valinomycin diffusion potential. The experiment comes with several advantages:

a.) as $\Delta\psi$ is the driving force, no opposing potential is formed

b.) no background signal is produced when the reaction is started conveniently by addition of valinomycin

c.) the direction of transport can be reversed depending on the potassium content of buffer and liposomes.

3. It is unfortunate that the rate of proton transport is not properly resolved. I suggest the following alternatives:

- use a stopped flow apparatus to increase time resolution. The pyranine signal can also just be followed at 460/510 nm, which eliminates the need of using slow ratiometric measurements.

-Instead of adding concentrated base (NaOH) or acid, the liposomes could also preincubated in an acidic buffer (e.g. 5.5) for some time, and rapidly diluted into buffer at higher pH values. This omits the sometimes-critical step of adding concentrated solutions that might trigger local effects on part the liposome population. These experiments are well established and are also compatible with stopped-flow measurements, where similar amounts of solutions need to be mixed. See (<https://www.sciencedirect.com/science/article/pii/S0014579399010601> or ref. 49).

- decrease the number of Nqo molecules in the liposomes. Taking the information from the methods section (200 nm liposomes, 300:1 lipid to protein ratio, $M_w \sim 80$ kDa, I calculated approximately 10 molecules per liposomes are used. This number can further be decreased, which would slow the transport and making it better detectable.

- Increase the buffering capacity of liposomes, e.g. using e.g 20 mM buffer instead of 2 mM buffer

4. As discussed in 1, if the presence of valinomycin makes no difference, this either indicated

electroneutral proton transport by Nqo or leaky liposomes. Unfortunately, polar lipid extract, as it has been used in the pyranine measurements, has a bad reputation to make very leaky liposomes (consult Figure S1 from Biochemistry 2012, 51, 8, 1577–1585). In addition, the soybean extract II-s used in the ATP synthase measurements are also considered to produce relatively leaky liposomes. Treatment with ether or acetone might help to remove contaminants to provide charge transfer. Tighter liposomes are typically made from synthetic lipids, either DOPC/E/G or even better POPC/E/G (ref) as the authors have used in their simulation studies. This is also important in respect to their measurement temperature of 37°C (probably because of of the *Thermus thermophilus* enzyme), which also make liposomes also leakier compared to measurements at room temperature.

5. Would it be possible to include a mutant variant that is expected to fully block transport? That would validate the relative findings of the presented mutant variants.

6. As mentioned in 3, the number of Nqo molecules is expected to have a linear impact on the observed signals. In that respect, it is critical to verify that similar amounts of protein is reconstituted into liposomes and is not affected by the mutation. As a worst case scenario, it could be envisaged that replacement of the charged glutamate by a neutral glutamine could have a beneficial effect on membrane protein integration in the absence of its ion salt partner. An increased incorporation would result in increased proton transport as observed, however without necessarily changing the electrostatics of the proton transfer as suggested.

On the other hand, I would not expect that protein orientation matters in this case as antiporter like systems often work in either direction.

Answer to comments by Reviewer #1 (Remarks to the Author):

Comment:

The work entitled “ Dissected Antiporter Modules Establish Minimal Proton-Conduction Elements in the Respiratory Complex I” presents a combined computational, using MD and QM/MM, and experimental study of a minimal proton-conducting membrane modules, created by the authors by engineering and dissecting the key elements of the bacterial Complex I, the highly intricate redox-driven proton pump that powers oxidative phosphorylation across all domains of life. In the work, the authors show that the isolated antiporter-like modules are able to conduct protons across proteoliposome membranes, and present evidence that the rate of proton conduction is controlled by conformational changes of buried ion-pairs that modulate the reaction barriers by electric field effects.

Overall the work is well done, figures are impressive and the researchers have a long tradition and experience in working with complex I, which is a relevant topic. The results are reasonably well supported by the presented data. In summary the relevance and quality of the work supports its publication. However, the work lacks in clarity and novelty, and therefore my overall feeling is that it could be better suited for a more specialized journal. Specific comments are described below. The two main issues with the present work are 1) novelty, and 2) clarity for the general readership.

Answer: We thank the Reviewer for the excellent comments, and for supporting the publication of our work. We appreciate the detailed suggestions that have helped us to further improve our work. In this regard, we have reformulated some sections of the manuscript to make it more accessible for the general reader, with answers to the comments addressed below.

We would like to emphasize that our work addressed the most hotly debated question in the field on whether all antiporter modules of Complex I transport protons across the membrane, or if this activity arises only from the terminal subunit, as suggested in recent Nature (Kravchuck *et al.* 2022) and Science (Kampjut & Sazanov, 2020) publications. We have indeed proposed a detailed mechanistic model, but experimental validation (of our and other mechanistic proposals) is still missing. The main reason is that mutations lead to a complete inhibition of the coupled proton and electron transport activities in the intact Complex I.

To this end, our work provides the first experimental evidence that the individual antiporter modules are indeed responsible for the proton pumping across the membrane, and that the proton transport is modulated by the ion-pair elements. These aspects have now been further clarified in the revised manuscript:

"The long-range energy transduction mechanism of Complex I has been of major interest for the last decades, with early proposals highlighting possible electrostatic and/or conformational effects involved in the process^{4,10,23,84,91} (but cf. also Refs.^{1,5,20,21,23,29,30,37} for detailed molecular mechanisms)."

[...]

"However, the direct biochemical characterisation of individual residues or even subunits involved in the proton transport process has so far been unsuccessful due to the tightly coupled proton- and electron transfer process, where mutations of key residues in the proton channels inhibit the Q-oxidoreduction activity, and vice versa.¹⁰⁻¹⁷ Therefore, despite the detailed mechanistic proposals,^{1,19,20,37} there is currently no experimental validation of the principles

underlying the proton transport in Complex I, which has thus remained one of the most controversial and debated questions in the field."

[...]

*"To derive a bottom-up understanding of the intricate proton pumping process, we design here minimal proton-conducting modules of Complex I that allow us **for the first time** to probe **the molecular** principles underlying the putative 'conductive' states during turnover **within the smallest proton-conducting unit of Complex I.**"*

[...]

"On a general level, the established integrative approach provides a systematic basis to study the function of individual units in complex multi-subunit membrane complexes, as well as an approach to probe, e.g., how disease-related mutations affect the individual proton transport reactions in Complex I."

Addition in the Abstract:

"Our findings provide direct experimental evidence that the individual antiporter modules are responsible for the proton transport activity of Complex I. "

Question:

1) The work main aim is to provide evidence supporting that the proton pumping is catalyzed by the antiporter-like subunits Nqo12, Nqo13, and Nqo14, together with Nqo7, through an S-shaped water mediate proton wire across the membrane, which, already in previous works were modulated by the conformational state of the conserved Glu-Lys ion pairs located at the interface of each antiporter-like subunit. Moreover, the proposal that the long-range proton transport process takes place by an electric wave that propagates in forward and reverse directions across the 200 Å wide membrane domain, similar to an 'electrical cradle' was also proposed in previous works. Since the role of the S haped tunnel, the ion pair and the electric wave were already proposed, and supported in previous studies it is not clear, beyond providing support for them what is the general novelty of the present work.

Answer: We have now better emphasised in the revised manuscript that our work provides the first experimental evidence that the individual antiporter modules of Complex I are responsible for the proton transport activity (see answer above). All previous attempts to study the ion transport activity in the antiporter modules have failed, while the mutagenesis in the complete intact Complex I is hampered by the inhibition of the tightly coupled proton-electron transfer reactions. There has been a long-standing interest in the field for these questions, but all previous attempts to characterize the elementary proton transport activities have been unsuccessful.

This was possible only by our long-term design of new genetic and molecular biological approaches that allowed the expression and purification of the constructs, together with the integration of biophysical experiments and molecular simulations.

As now also better clarified in the revised manuscript, there are several recent conflicting proposals regarding the location of the proton pathways, and the mechanistic models that have been difficult to test experimentally. Importantly, the idea that each antiporter module pumps protons was challenged by the recent suggestion that all protons are translocated by the terminal (Nqo12) subunit. Moreover, the mechanistic principles underlying both our

electrostatic wave mechanism as well as other proposals have remained much debated due to the difficulty to experimentally test and validate or invalidate these models due to the strong coupling between the proton transfer and the Q oxidoreductase activity.

Our work is highly significant and novel, as we can now experimentally show that the different antiporter modules indeed transport protons across the membrane, as predicted by our multiscale computational predictions. In this regard, by dissecting the minimal proton conducting elements, we can address the exact functional elements responsible for this activity from a bottom-up approach. Our work indeed supports key features of the electrical cradle mechanism (Kaila 2018 Roy. Soc. Interface), where we propose that the ion-pair elements modulate the proton transfer reactions in each of the antiporter-like subunits.

Revisions in the main text:

*"To derive a bottom-up understanding of the intricate proton pumping process, we design here minimal proton-conducting modules of Complex I that allow us **for the first time** to probe **the molecular** principles underlying the putative 'conductive' states during turnover **within the smallest proton-conducting unit of Complex I**. Our constructs are built based on multi-scale simulations, and probed experimentally by biophysical proton conduction experiments in proteoliposomes in combination with mutagenesis studies, **with optimisation of new genetic and molecular biological approaches that allowed the expression, purification, and characterization of the engineered constructs (Supplementary Fig. 10, Supplementary Table 10)**. Our combined results show that the antiporter-like subunits Nqo13 (ND4) and Nqo12 (ND5) contain all functional motifs required to transport protons across membranes, and that the buried ion-pairs of the antiporter modules form key elements that control the rate of proton conduction. **On a general level, the established integrative approach provides a systematic basis to study the function of individual units in complex multi-subunit membrane complexes, as well as an approach to probe, e.g., how disease-related mutations affect the individual proton transport reactions in Complex I.**"*

Question:

2) The work is very hard to follow for general readers which are not specialists in Complex I structural biochemistry. Several key aspects are not explained, and should be. For example, i) how did the authors come to the hypothesis of the key ion pair as being the controller of the process. ii) The hypothesis of the propagating electric wave should be explained.

Answer: We thank the Reviewer for this valuable remark. We have now re-written parts of the more specialised sections to make them more accessible for the general reader. Regarding the ion-pair hypothesis, we clarify references to our previous work, in which we proposed that the conformation of the ion-pairs modulates the proton transfer barrier in the antiporter modules (Di Luca 2017, Kaila 2018, Mühlbauer 2020, Röpke 2020). We now also better explain how the mutations were designed in our current work.

Revisions in the main text:

*"Previous work^{1, 13, 30, 31, 37} also found that the proton transfer barrier is modulated by the conformational state of a conserved Glu-Lys ion-pair, located at the interface of each antiporter-like subunit (Fig. 1b). **More specifically, opening of the conserved ion-pair was found to lower the barrier of lateral proton transport within an antiporter subunit, and in turn, favour the conformational changes in the ion-pair of the neighbouring subunit, leading to the subsequent propagation of a protonation reaction (cf. Refs.^{1,30,37}).**"*

“In order to induce dissociation of the ion-pair, mimicking a conductive state along the wave propagation model^{1, 13, 37}, we replaced Glu123 (Glu132 in Nqo12) by a glutamine residue in the simulation models (Fig. 2b).”

The long-range proton transport process was suggested¹ to take place by an electric wave that propagates in forward and reverse directions across the 200 Å wide membrane domain, similar to an 'electrical cradle' (cf. Newtonian cradle^{1, 13, 37}), leading to the release and subsequent uptake of protons by conformational changes in the ion-pairs.

“In addition to their key role in the antiporter modules of Complex I, buried ion-pairs establish functional motifs also in several other enzymes...”

Question:

Concerning the methodology, I will focus on computer simulations 3) The authors computed using QM/MM a minimum energy reaction pathway (MEP) followed by umbrella sampling the proton transfer reaction. For this reaction, the reaction coordinate is never presented nor explained. More important, if I understood correctly only one step of proton transfer was studied, and not the whole path, or at least all key proton transfer steps. In this sense the works seems to be incomplete.

Answer: We thank the Reviewer for pointing out that a more detailed description of the methodology was needed. The reaction coordinate for the lateral proton transfer reactions from middle Lys to terminal Lys/Glu, was modelled as a linear combination of bond-breaking and bond-forming distances. The definition of the reaction coordination is now shown in the Supplementary Figure 6:

We have also clarified that the focus of these QM/MM free energy simulations is on the lateral proton pathways as these are close to the proposed gating elements and introduced substitutions that we test and validate here experimentally. We would like to emphasize that the QM/MM free energy simulations are computationally highly challenging, and the 10 Å long lateral water wires already push the sampling limits. A similar treatment of ca. 50 Å long proton transfer reactions is thus outside the scope of the work.

Revisions in the Main text:

“To estimate the proton transfer energetics along the *water-mediated proton wires*, we next employed hybrid quantum/classical (QM/MM) free energy calculations performed based

on the microsecond MD simulations that allowed us to capture the energetics for the proton transfer along the lateral pathways (Fig. 2i, Supplementary Fig. 6). These lateral proton pathways were the main focus here due to their proximity of the ion-pairs and the central proton transfer elements (but see below). For Nqo13 structures with a closed ion-pair conformation, we obtain a proton transfer barrier between Lys235 and Glu377 of $\Delta G^\ddagger \sim 14$ kcal mol⁻¹ and an overall $\Delta G \sim +7$ kcal mol⁻¹ (Fig. 2i), whilst dissociation of the (Lys204-Glu123) ion-pair drastically lowers the proton transfer barrier by electrostatic effects (Fig. 2j, see below). Although the complete free energy exploration of the ca. 50 Å long proton transfer pathway is not technically feasible with the current methods, we note that the proton uptake to the middle lysine is not rate-limiting for the overall process, with a rather low reaction barrier ($\Delta G^\ddagger < 5$ kcal mol⁻¹, see Methods, Supplementary Fig. 6)."

Revisions in Methods:

"To estimate the free energy landscape of the proton uptake from N-side (Lys282 to Lys235 in Nqo13), we used a QM/MM simulation together with the recent shared-bias well-tempered metadynamics-extended adaptive biasing force (MWE) method^{86,87}. Here, the reaction phase space was explored by a harmonic potential $B(\zeta(x), \lambda)$, coupled to a moving fictitious particle λ , with an additional time-dependent bias potential $B_{MWE}(\lambda, t)$ used to ensure a uniform sampling. Parallel sampling was achieved by performing the simulation with 10 walkers (bin width=0.1 Å), each sampled for 22.5 ps with a 0.5 fs timestep and at T=310 K. Gaussian hills with heights and widths of 1.0 kJ mol⁻¹ and 0.2 Å, respectively, were added every 20 fs. The QM region comprised ca. 150 atoms (see Supplementary Table S9). The reaction coordinate (CV) was defined using the modified centre of excess charge (mCEC), with the sidechain nitrogen atom of the residues Lys282 (N-side) and Lys235 (middle Lys) set to donor and acceptor, respectively, spanning the CV range [0, 10 Å]. The QM/MM-MWE-sampling was performed using OpenMM/Fermions++⁸⁸ with the unbiased PMF retrieved by MBAR⁸⁹ analysis."

86. Pöverlein M. C., Hulm A., Dietschreit J. C. B., Kussmann J., Ochsenfeld C., Kaila V. R. I. QM/MM Free Energy Calculations of Long-Range Biological Protonation Dynamics by Adaptive and Focused Sampling. *J. Chem. Theory Comput.* (In Press).

87. Fu, H., Shao, X., Cai, W., Chipot, C. Taming Rugged Free Energy Landscapes Using an Average Force. *Acc. Chem. Res.* **52**, 3254–3264 (2019).

88. Kussmann J., Ochsenfeld C. Pre-selective screening for matrix elements in linear-scaling exact exchange calculations. *J. Chem. Phys.* **138**, 134114 (2013)

Using a new QM/MM free energy method (Pöverlein *et al.* JCTC in press, Ref. 86), we could further show that the proton transfer from the N-side to the middle Lys is not rate-limiting for the overall process. The proton release step to the P-side may involve higher barriers, but these are further away (>25 Å) from the introduced substitutions and thus unlikely to modulate the reaction barriers. Treatment of the complete free energy landscape for the proton transfer reaction requires development of new methodology, which will be the focus of a future work.

Supplementary Fig. 6 [...]. i) The free energy of proton uptake from the N-side to the middle Lys235 in Nqo13, using the modified centre of excess charge (mCEC) as a reaction coordinate / collective variable (CV (Å)), using the modified centre of excess charge (mCEC) as a reaction coordinate / collective variable (CV (Å)), see *Methods*. **j)** Convergence of the MWE-QM/MM free energy profiles with increasing sampling time. **k)** Snapshots along the N-side proton transfer reaction at different CVs, from the N-side (CV = 0 Å) to the middle Lys235 (CV = 11 Å).

Question:

4) Concerning the classical MD simulations, and the observed hydration of the S shaped proton channel. What I feel is missing is an evaluation of the pKa of key residues along the channel, how the change in their protonation state results in small conformational changes (i.e side chain movements), and how this changes due to the electric wave propagation model, or more general the effect of ET from NADH to Q.

Answer:

The pKa values of titratable residues along the proton pathways are shown in the Supplementary Table 3, and Supplementary Figure 16. In this analysis, we compare the pKa

shifts in the antiporter modules to the intact Complex I. To further address how conformational changes could modulate these pK_a shifts, we employed the constant pH-MD simulation approach that allowed us to estimate the shifts due to the conformational changes as well as residue substitutions. In the discussion section we have placed these pK_a shifts in context of our electric wave propagation model and discussed how these could be triggered by the Q oxidoreduction activity.

Supplementary Table 3 | Predicted pK_a s in the dissected antiporter-like subunit relative to the intact Complex I. The pK_a computed based on PDB ID:6I0D⁶, and ΔpK_a s of the isolated Nqo13 / Nqo12 constructs relative to WT intact CI based on MD ensemble (see *Extended methods*).

Nqo13	pK_a X-ray, CI full	pK_a X-ray, 13 only	ΔpK_a X-ray CI full - 13 only	ΔpK_a MD: CI full - WT/Nqo13	ΔpK_a MD: CI full - E123Q/Nqo13	$\Delta\Delta pK_a$ MD: WT - E123Q
K282	15.0	15.8	-0.8	-4.1	0.4	+4.4
D228	-4.7	-3.8	-0.9	-2.6	0.5	+3.1
H218	-0.7	0.5	-1.2	-4.8	2.2	+7.0
K235	8.6	9.8	-1.2	-2.4	-1.3	+1.1
H292	2.9	6.8	-3.9	-2.5	-1.9	+0.6
E377	5.8	4.7	+1.1	-5.3	-1.6	+3.7

Nqo12	pK_a X-ray, CI full	pK_a X-ray, 12 only	ΔpK_a X-ray CI full - 12 only	ΔpK_a MD: CI full - WT/Nqo12	ΔpK_a MD: CI full - E132Q/Nqo12	$\Delta\Delta pK_a$ MD: WT - E132Q
E346	4.4	4.6	-0.8	-0.1	+0.6	+0.7
K292	10.5	10.8	-0.3	+0.6	+0.4	-0.2
H241	-1.2	-0.6	-0.6	-3.8	-0.9	+2.9
K329	9.7	10.1	-0.4	-2.2	+1.8	+4
H325	-4.3	-4.1	-0.2	+0.0	+2.8	+2.8
H321	2.6	2.9	-0.3	+2.3	+3.4	+1.1
K385	13.5	13.7	-0.2	+4.7	+6.7	+2.0
D386	5.4	6.2	-0.8	-2.0	-0.4	+1.6

" with mutations in ion-pairs (e.g. E123Q-Nqo13) inducing conformational changes and pK_a shifts as also supported by our constant pH-MD simulations (Supplementary Fig. 16)."

Supplementary Fig. 16 (contd.) | Analysis of pK_a tuning effects in the dissected antiporter-like subunit and the intact Complex I. The figure shows the protonation probability ($\langle x \rangle$) as a function of pH. c) Titration curves of residues along the proton pathway for the Nqo13 WT (top), and Nqo13 E123Q constructs, based on constant-pH MD simulations (see *Extended Methods*).

Extended Discussion

Conformational changes coupled to proton transport

To test how the proton transfer reactions coupled to conformational changes in the antiporter-like subunits, we performed additional MD simulations in multiple states. In the WT-Nqo13, we find that upon transfer of the proton from Lys235 to Glu377, mimicking a proton transfer step along the central pathway, the sidechain of His292 flips to a different rotameric state, suggesting that the conformation of His292 could affect the back transfer of protons in the reverse direction (*cf.* also Refs.^{13,14}, Supplementary Fig. 1g). Moreover, we note that the deprotonation of the Lys235 leads conformational changes in conserved residues in TM7a/b and TM8 (Supplementary Fig. 1h) that could regulate the proton uptake from the N-side bulk. The constant pH-MD simulations (see *Extended Methods*) further support conformational changes around the broken TM7a/b helix, suggesting that the effective pK_a of Lys235 is around 9.3 in the WT-Nqo13, and shifts to 6.5 for E132Q-Nqo13 (Supplementary Fig. 16).

Extended Methods

Constant-pH MD simulations

Constant-pH MD (cphmd) simulations were performed based on the 1 μ s MD simulations of Nqo13 embedded in a POPC membrane and solvated in a water box (see main text *Methods*). The cphmd simulations were performed using 15 independent simulations, with 20 replicas each, sampling a pH range from 0.5 to 13.5, and with initial protonation states assigned from the PBE/MC calculations (see above). Each simulation included 1000

minimisation steps, followed by 500 cycles of 1000-10000 production steps and 7500 non-equilibrium switching steps, leading to a total simulation length of *ca.* 50 ns. Final pK_a values were computed based on all sampled conformations using the PBE/MC methodology (see above) by re-weighting each conformation by its protonation probability and electrostatic energy to derive average protonation states. The cphmd simulations were performed, as implemented in NAMD¹⁰⁻¹², and pK_a calculations using an in-house APBS/Karlsberg+ implementation²⁻⁵.

Question: ... and how this changes due to the electric wave propagation model, or more general the effect of ET from NADH to Q.

Answer:

The general mechanistic process is explained here:

"The proton pumping in Complex I could be initiated by conformational changes in charged networks triggered by the movement of the quinol species from the hydrophilic domain to a second membrane-bound Q binding site^{1, 9, 22, 33} that in turn triggers lateral proton transfer reactions towards the Nqo11/Nqo14 interface^{35, 36} (Fig. 5b). This protonation cascade leads to stepwise opening of ion-pairs that favour the lateral proton transport towards the next antiporter, and leading to the propagation of 'protonation signal' via Nqo14 and Nqo13 to Nqo12. The well-wired P-side connections in Nqo12^{19, 20, 28, 30, 34} could initiate proton release across the membrane (see above). The proton release increases the proton affinity of the "middle Lys", favouring closing of the ion-pair at the Nqo12/Nqo13 interface, and releases the "Nqo13 proton" across the membrane. Re-protonation of Lys235 and closing of the Glu123-Lys204 ion-pair similarly eject the "Nqo14 proton" across the membrane, followed by propagation of the signal to the membrane-bound Q-binding site. This re-sets the machinery by releasing the quinol and leading to uptake of a new quinone from the membrane pool. The proton transport in the antiporter modules supports that the ion-pairs modulate the proton transfer barriers, as well as central residues along the putative proton pathways. Taken together, we suggest that the isolated antiporters also utilise similar conformational changes in the ion-pairs and wetting/drying transition to channel the protons across the membrane (Fig. 5a). "

Answer to comments by Reviewer #2 (Remarks to the Author):

Comment:

The paper explores the properties of two membrane subunits, Nqo 12 and 13, in respiratory Complex I, responsible for transmembrane proton transfer. From the finding that these isolated and reconstituted subunits increase the membrane proton permeability, the authors suggest that these isolated and reconstituted subunits comprise all necessary functional elements for conducting protons across membranes. However, the conclusion and experimental approach appear to lack logical coherence.

Given that the studied subunits are exposed to high electrochemical proton potential in situ (up to 200 mV in mitochondria), they should not induce proton leakage, as this would establish a futile cycle. Proton channels in Nqo12 and 13 should open only in response to the wave of electrostatic interactions initiated by quinone reduction. The authors acknowledge this in the discussion, and to overcome this problem they propose that in vivo assembly proteins prevent proton dissipation by blocking native antiporter contacts. Isolated and reconstituted without assembly proteins, Nqo12 and 13 might produce proton leaks not through their channels but due to the loss of correct surrounding or folding.

Wild-type Nqo12 and 13 exhibit proton permeability slightly higher than the natural permeability of the liposome membrane. Wild-type Nqo12 and 13 exhibit proton permeability slightly greater than the natural permeability of the liposomal membrane. Mutations leading to conformational changes lead to a significant increase in proton fluxes; according to the authors, this indicates the control of proton channel by replaced amino acid residues. However, there is no evidence whether proton flux is mediated by a natural proton translocation mechanism or is due to destabilization of the lipid environment by a misfolded protein creating an artificial proton pathway.

Answer:

We thank the Reviewer for the insightful comments and suggestions that have helped us to improve our manuscript. To show that the proton conduction activity does not arise from unfolding of the antiporter modules, we studied each construct (WT and the designed mutations) by GFP fusion-based fluorescence-detection/size-exclusion chromatography (FSEC) experiments. To this end, we find that all samples elute similarly, indicating that all constructs have overall similar physico-chemical properties regardless of the mutation, whilst unfolding would lead to a different elution profile, and cleavage of the GFP. In this regard, we noted that only the H292A¹³ construct behaved differently from the other constructs, and leading to cleavage of the GFP-fusion protein in some sample preparations. To this end, we decided to remove the description of the H292A¹³ constructs. We further note that the proton transport activity cannot arise from unfolding as the integrity of the liposome membrane is perfectly conserved, and we were able to design three mutants that completely block the proton conduction activity in the studied conditions.

Revisions in the main text / SI:

"Our GFP fusion-based fluorescence-detection/size-exclusion chromatography (FSEC)⁴³ further supported that all expressed and purified protein constructs were well-folded (Supplementary Fig. 10j,k)."

Supplementary Fig. 10 | Construct design, expression, purification, and stability of dissected antiporter-like modules. **a**) Nqo12^{ATH} expression trial using BL21 strain followed by GFP in-cell fluorescence. **b**) In-gel fluorescence of Nqo12^{ATH}-sfGFP (without linker) monitoring IMAC fractions. **c**) Size-exclusion chromatography profile showing the absorbance at 280 nm and GFP excitation at 488 nm of Nqo12^{ATH}-sfGFP without linker expressed in BL21. **d**) Size-exclusion chromatography profiles showing the absorbance at 280 nm and GFP excitation at 488 nm. **e**) In-gel fluorescence and **f**) Coomassie gel of Nqo12^{ATH} following the purification fractions (CL: cell lysate, MBp: membranes pellet, MBs: membrane centrifugation supernatant, FT: flow through, W: wash, E₁: IMAC elution, E₂: SEC elution). **g**) Nqo orientation in liposomes measured using the NTA-Atto 647 N fluorescent label. **h**) In-gel fluorescence image and **i**) Coomassie gel of Nqo13 following the purification fractions (CL: cell lysate, MBp: membranes pellet, MBs: membrane centrifugation supernatant, FT: flow through, W₁₋₂₋₃: washes, E₁: IMAC elution, E₂: SEC elution). SEC profiles for **j**) Nqo12 and **k**) Nqo13, eluted at 60 mL, showing GFP fluorescence followed at 488 nm. **l**) Gel showing sfGFP removal by overnight cleavage with TEV protease. Growth tests performed in TB and minimum medium (M9) for **m**) Nqo12 and **n**) Nqo13 in induced (I) and non-induced (NI) conditions. **o**) In-gel fluorescence assessing the reconstitution efficiency of the Nqos into proteoliposomes.

Comment/question:

Regarding the results, there are several comments on the treatment of biochemical data. The initial phase in Fig 3b is not resolved.

A major part of this phase is formed by the response of not inner, but outer pyranine since the removal of the outer dye never can be 100%. This means that the initial phase of the proton flux is lost and 5-digit numbers of k_2 (Supplementary Table 1) do not have any meaning.

Answer:

We thank the Reviewer for this comment and fully agree on this point. We have now made significant technical improvements in our spectroscopic experiments, now allowing for data collection with a much higher time-resolution than before.

However, for the acid-induced experiments, we now focus on the steady-state conduction, and quantify the steady state ΔpH created by antiporter modules, which provides a more robust approach, but nevertheless fully supports the kinetics characterized in the independent ATPase assays. We have also reduced the number of significant digits in the data fitting.

"The time-resolution of the ACMA fluorescence experiments is 100 ms, and 600 ms for the pyranine experiments."

	ATP synthase					
	Linear fit					
Construct	avg. k_1	SD	n	Plateau ΔF^*	SD	n
EL	0.04	0.02	6	0.96	0.01	6
AqpZ	4.42	0.49	6	0.09	0.01	6
F ₁ F _o	5.56	0.40	6	0.07	0.01	6
WT ¹²	2.53	0.28	6	0.33	0.04	6
E132Q ¹²	0.58	0.18	6	0.71	0.06	6
K385I ¹²	4.98	0.19	6	0.13	0.02	6
WT ¹³	2.36	0.19	6	0.32	0.03	6
E123Q ¹³	2.00	0.27	6	0.43	0.04	6
K235M ¹³	2.72	0.28	5	0.31	0.03	5

*Extrapolated from a single exponential fit.

Question:

The titration curve of pyranine with a pKa over 8 (Supplementary Fig11a), inconsistent with the known pKa of 7.2, raises questions.

Answer:

We have now clarified that our experiments are performed at 37°C as well as in buffer conditions, which are known to affect the pKa of pyranine (for example Fig. 1D in <https://www.mdpi.com/1420-3049/27/8/2490>).

Revisions in the SI text:

"The higher measuring temperature (37°C) as well as the buffer conditions lead to an upshift in the pyranine pKa from 7.3 (cf. e.g. Ref¹⁵ for exploration of buffer effects)."

15. Scarciglia A, Di Gregorio E, Aime S, Ferrauto G. Effects of Cations on HPTS Fluorescence and Quantification of Free Gadolinium Ions in Solution; Assessment of Intracellular Release of Gd³⁺ from Gd-Based MRI Contrast Agents. *Molecules*. **27**, 2490 (2022).

Question:

The fitting of curves is compromised by poor time resolution, with initial experimental points often deviating from fitted curves. However, some experimental results, such as the addition of NaOH, methylamine, and acetate, seem excessive and do not provide new information beyond what is demonstrated with co-reconstitution with the slow-operating ATPase. These experiments show quite clearly that Nqo12 and 13 cause some proton leakage; the more these proteins are disrupted by mutations, the greater the leakage of protons.

Answer:

We thank the Reviewer for the suggestion. We have now improved the time resolution of the data collection, together with re-optimisation of antiporter modules in the proteoliposomes that improved the signal-to-noise ratio. Although we argue that our combined experiments provide important independent validation, we also agree with the Reviewer that some experiments were excessive. In this regard, we have decided to remove the description of the NaOH conditions.

Question:

In the introduction, the mention of long-range proton transfer should reference the concept of the electrostatic interactions wave in Complex I, first suggested in 2008 by Euro et al. (BBA 1777, p1166) and described in more detail in 2013 by Verkhovskaya and Bloch (Int J Biochem Cell Biol. 45(2): 491).

Answer:

There are several early mechanistic ideas, but here we wanted to limit the discussion to make our work more accessible for the general reader, as covering these will require a detailed description. It is however, important, not to mix the proposal of Euro et al (2008) that pre-dates the resolved structures of Complex I or the "wave-spring" proposal of Verkhovskaya and Bloch with our electrical wave mechanism, which operates by different principles and includes exact molecular details that form the basis of our study. Following the Reviewer's suggestion on crediting early mechanistic ideas we now write in the main text:

"The long-range energy transduction mechanism of Complex I has been of major interest for the last decades, with early proposals highlighting possible electrostatic and/or conformational effects involved in the process^{4,10,23,84,91} (but cf. also Refs.^{1,5,20,21,23,29,30,37} for detailed molecular mechanisms)."

Comment:

In summary, the paper is overly verbose, containing excessive data and an overload of figures and numbers in the biochemical part. Streamlining the content and focusing on key findings could enhance clarity and impact.

Answer:

We thank the Reviewer for the constructive comments and suggestions. We have followed the Reviewer's advice that has made our manuscript more accessible for the general reader and clarified the impact of our work.

Answer to comments by Reviewer #3 (Remarks to the Author):

Comment:

Begiah et al describe their ambitious efforts to deconstruct a bacterial complex I (Thermus thermophilus) into parts and check for their remaining function. The focus is on the two membrane embedded antiporter like modules nqo12 and nqo13, which they successfully express in E. coli and purify via affinity chromatography. To facilitate expression, they remove the transversal helix from nqo12 and they can convincingly show that they have expressed and purified the two proteins.

Answer:

We thank the Reviewer for appreciating the challenges and impact of our work, and for the detailed insightful suggestions that helped us further improve our manuscript. We have addressed each specific question in our *point-by-point* answer below.

Comment:

To get an idea how these proteins could behave isolated in membranes, they perform molecular dynamics simulation studies of the individual subunits in lipid bilayers, either in pure POPC or the more natural variant POPE/POPG/CDL. While they show the simulation data, they unfortunately do not discuss if the observed differences are relevant, or if the choice of lipids does not make a big difference. For their liposome assays, they settle for E. coli polar lipid extract, which is closer to second setting (see more on this below).

Answer:

We thank the Reviewer for the suggestion. In our MD simulations, we indeed tested both pure POPC lipid models, as well as three-component lipid membranes consistent with the *E. coli* polar lipid composition that we used also experimentally. Although the focus of the simulations was not on the protein-lipid interactions, we agree with the Reviewer that these are interesting and important effects. To this end, we expanded the simulation and analysis of the lipid effects. Interestingly, we find that the protein-lipid interface is more sealed in the constructs embedded in polar lipid membrane composition, but the protein conformational dynamics is similar between our two models. This indicates that the possible non-specific leak reactions at the protein-lipid interface are not at least enhanced by the polar lipids, which are otherwise known to be experimentally leakier relative to pure lipids.

Revisions in the main text / SI:

*“During the microsecond molecular dynamics simulations, the isolated antiporter modules remain stable (Supplementary Fig. 1d-f), and dynamically closely resemble the analogous subunits within the intact Complex I simulations (Fig. 1a, b, 2a, b, Supplementary Fig. 1d-f)^{1, 13, 28-31}. **Particularly, cardiolipin of the polar lipid model establishes tight contacts with the antiporter modules and seals them from the bulk phase (Supplementary Fig. 4)***

Supplementary Fig. 4 | Effect of the membrane composition and protein-lipid interactions. **a, b)** Hydration along the proton pathways in **(a)** Nqo13 and **(b)** Nqo12^{ΔTH} constructs in a pure POPC membrane and in a PE:PG:CDL mixture for the WT (water molecules in blue) and ion-pair mutant (water molecules in orange) simulations. **c, d)** Cardiolipin (CDL) molecules stabilise the protein-lipid interface in the MD simulations with the PE:PG:CDL membrane for **c,** Nqo13 and **d,** Nqo12^{ΔTH}.

Question:

They identify several residues that are expected to influence the activity of the proton path, namely the intersubunit bridging residues (via salt bridge) or polar residues along the proton paths. No further explanation is given on which basis the replaced amino acids were chosen. While an exchange of a glutamate with a glutamine (as in E123Q or E132Q) is not surprising, an exchange of a lysine with an isoleucine (as in K385I) deserves more attention as it might also affect the overall protein stability and functionality.

Answer:

We have now better justified how the mutations were designed and guided by our molecular simulations:

The E123Q¹³ and E132Q¹² constructs were designed to mimic an open ion-pair conformation, by replacing the anionic residue (Glu) by a neutral polar residue (Gln), leading to a smaller electrostatic attraction with the nearby lysine. As explained in the main text, this mutation was also designed to mimic the effect of protonation of the Glu, that could occur in the full complex I if the neighbouring antiporter-like subunit transfers a proton from the terminal site, as

proposed before (Röpke, JACS). A similar logic was used to replace other residues along the putative proton pathways (e.g. E377Q¹³). Moreover, to create blocking mutants (see below), we removed the middle or terminal proton donor/acceptors along the proton pathways as identified in the MD simulations. In this regard, we replaced the residues with Met or Ile residues (K235M¹³ and K385I¹² constructs), as they have a similar volume to the Lys residue, and indeed lead to a non-polar sealing of the proton pathway as suggested by our MD simulations. Despite the different physico-chemical properties of some introduced residues, we show that the protein stability is maintained in all constructs (see below).

Revisions in the main text / SI:

"In order to induce dissociation of the ion-pair, mimicking a conductive state along the wave propagation model^{1, 13, 37}, we replaced Glu123 (Glu132 in Nqo12) by a glutamine residue [...]."

"In order to block the possible proton pathways, we replaced the middle lysine in Nqo13 and terminal lysine in Nqo12 with Met or Ile residues (K235M¹³ and K385I¹²), due to their similar volume but non-polar character. Our MD simulations of these constructs suggest that these substitutions could indeed lead to a partial sealing of the proton pathway (Supplementary Fig. 2f, 3d)."

"Our GFP fusion-based fluorescence-detection/size-exclusion chromatography (FSEC)⁴³ further supported that all expressed and purified protein constructs were well-folded (Supplementary Fig. 10j,k)."

Question:

In their main experimental part, they perform three types of proton conducting measurements across the membrane.

1. The establish a ΔpH by adding base NaOH to the outside of the liposome, thereby creating an driving force for outwards proton transport, which they follow using the pH sensitive dye pyranine entrapped in the liposomes. In their liposomes they reconstitute either different Nqo variants, or AqpZ and no protein as negative controls. Throughout their experiments, no difference between empty liposomes or liposomes containing AqpZ are observed.

While they see differences between the negative controls and their Nqo variants, they majority of the signal seems unspecific, making interpretation difficult. They observed a biphasic behaviour of the transport, however they do not comment on the relevance or origin of the slower transport rate. The display of the data is somewhat unfortunate, as the kinetics of the rapid (but more relevant and later discussed) phase is not resolved. The author might consider using a different form, e.g. broken x-axis or two graphs with different time scales.

Answer:

We thank the Reviewer for the insightful comments. Following the Reviewer's suggestion, we have improved the time-resolution of the data collection, together with re-optimization of antiporter modules in the proteoliposomes that allowed for a better *signal-to-noise* ratio. In the revised manuscript, we now also speculate that the slower biphasic rate could be caused by a slow non-specific proton diffusion through the liposome membrane. However, following the advice of another Reviewer, we have decided to remove excessive experimental data, and therefore not to include the NaOH conditions, which indeed had the worst *signal-to-noise* ratio. As discussed below, we have focused now on the steady state proton conduction levels, which removes some of the caveats discussed above.

"followed by a slower alkalisation possibly due to a slow non-specific leak reaction (Fig. 3b, c, Supplementary Table 1, Supplementary Fig. 11b, c)"

Question:

2. Next, they switch to an approach, in which they add potassium acetate to the liposome. The expectation is that acetate will be protonated to a small degree ($pK_a \sim 4.8$) at pH 7.2, and in its protonated form it will diffuse across the liposomal membrane. Once inside the liposome, the same dissociation distribution applies and most of the transported acetic acid will be present in its ionized form, releasing the proton and thus acidifying the liposomal lumen. They indeed see a rapid decrease in the lumen of the liposomes, and they find that this drop is less pronounced in liposomes containing Nqo compared to control liposomes (empty or AqpZ). The rationale is that Nqo allows for rapid backflow of protons and thus a decreased ΔpH is established.

In their supplementary figures, they show a similar experiment, in which they use methylamine ($pK_a \sim 10.6$) to establish an inverse pH gradient (inside basic) and again observe competitive Nqo proton transport. Again, the display of the data does not allow to resolve the kinetics and the conclusions are solely drawn on the amplitude of the total drop (i.e. the resulting ΔpH within the different variants). The difference between empty liposome and liposomes containing Nqo variants are small and hard to appreciate, but they behave in the expected manner.

Answer:

We thank the Reviewer for nicely summarizing our results, and highlighting both strengths and weakness in our experimental setup. To address these questions, we have improved the time resolution of the data collection together with re-optimisation of antiporter modules in the proteoliposomes that allowed us to obtain improved estimates of the differences between the constructs. However, for these weak-acid/base experiments we decided to focus on the steady state conduction, and quantify the ΔpH dissipated by antiporter modules. This approach better highlights the key differences between the constructs, but they also support the kinetics characterized in the independent ATPase assays.

Revised figures and main text:

Fig. 3. Biophysical characterisation of proton conduction in dissected antiporter-like modules induced by addition of weak acid. **a)** Proteoliposome assay for probing the proton conduction kinetics in the dissected antiporter-like subunits with pyranine (HPTS) by addition of $K^+CH_3COO^-$. Addition of 10 mM acetate to **b)** Nqo12^{ATH}- and **c)** Nqo13- proteoliposomes induced conduction of protons across the membrane. The ionophore nigericin was added to dissipate the generated ΔpH . **d, e)** The final steady-state ΔpH level before addition of nigericin is shown for the **d)** Nqo12^{ATH} and **e)** Nqo13 constructs. The protonation conduction rate is sensitive to substitution of the conserved ion-pair and residues along the proton pathway. The data is compared to proton conduction assayed with empty liposomes (EL) and aquaporin (AqpZ) reconstituted proteoliposome. See Supplementary Fig. 10 for addition of weak base, and Supplementary Fig. 13b,c for ΔpH and ΔpH -mediated proton transport.

"In stark contrast to the empty liposomes and AqpZ, addition of potassium acetate to the Nqo13 and Nqo12 proteoliposomes resulted in rapid Nqo-mediated proton transport (<600 ms, see Methods) across the proteoliposomes membrane ($CH_3COOH_{in} \rightarrow CH_3COO^-_{in} + H_{in}^+ \rightarrow H_{out}^+$, Fig. 4a) as indicated by the fast pH increase and a shifted equilibrium between the CH_3COO^- / CH_3COOH forms by dissipation of the Donnan potential (Fig. 4c, e). In this regard, the steady-state ΔpH level reaches 7.1-7.12 for the WT Nqo13 and Nqo12^{ATH} relative to the empty liposomes/AqpZ ($\Delta pH < 7.04$), while the introduction of the glutamine at the positions Glu123 (E123Q-Nqo13) or Glu132 (E132Q-Nqo12) strongly increases the proton conduction rates (Fig. 3b, c, Supplementary Table 1), and leading to a further shift in the steady-state ΔpH level to a pH of around 7.15, consistent with our computational predictions of the lowered proton transfer barrier (Fig. 2c-j). Moreover, our experiments show that proton conduction can also be blocked by substitutions of the key residues along the proton wire in the studied conditions, leading to the same steady-state ΔpH level and proton conduction kinetics as for the empty liposomes. In this regard, we find that the removal of the terminal Lys385 (K385I) in Nqo12^{ATH}, or replacement of the 'middle lysine (K235M) or the terminal glutamate (E377Q) in Nqo13 (Fig. 3c) also blocks the proton conduction, suggesting that the residues are important for establishing a tightly gated proton transport machinery in Complex I."

Question:

3. In a third approach, they use purified E. coli ATP synthase and co-reconstitute with the different variants. Instead of pyranine as used above in the two previous approaches, they use ACMA which is traditionally applied to monitor acidification of the luminal pH in liposome experiments, which is observed as a quench in fluorescence. They find less quench in liposomes containing Nqo upon addition of ATP compared to empty liposomes. The rationale is that the co-reconstituted Nqo prohibits the build-up of a large ΔpH by parallel extrusion of the ATP synthase pumped protons via the co-reconstituted Nqo. In contrast to the previous measurements, no "background signal is observed in the empty liposomes", giving clearer distinction between liposomes having proteins reconstituted. Unfortunately, only normalized data are shown and the actual extent of ACMA quench is not visible. The relative fluorescence goes to 0 (or even below in 4f, blue trace) which makes not much sense in the case in an ACMA experiment (in the reviewer's experience, even a perfect ACMA quench experiments leaves 5 to 10% fluorescence behind). I don't fully understand what is set as the lower value of the normalized fluorescence (The minimal value of wt trace????). That should be explained better and/or the data should be shown unprocessed.

Answer:

We thank the Reviewer for this important comment, which made us realize that our data analysis created some potential confusion. We have now clarified our data processing approach, and now show only the relative fluorescence, without normalization. Indeed, our assay shows that the F_1F_0 -ATPase activity leads to a ca. 90% quenching level, consistent with previous literature. To further optimise the sensitivity of our assays, we developed a LMNG-purification approach of ATP synthase. All data was re-measured using the better optimised condition. The figures have been updated and are shown below:

Fig. 4. Biophysical characterisation of proton conduction properties in dissected antiporter-like modules co-reconstituted with F₁F_o-ATP synthase. **a)** Proteoliposome assay for probing the proton conduction kinetics in the dissected antiporter-like subunits co-reconstituted with ATP synthase, monitored by fluorescence quenching of ACMA. Addition of 0.2 mM ATP generates an ATPase-driven Δ pH across the proteoliposome membrane, which competes with the **d)** Nqo12^{ATH}- and **f)** Nqo13-mediated proton transport. **b, c)** Structure of the proton pathways from MD simulations of the WT (blue) and ion-pair mutants (orange) for **b)** Nqo12^{ATH} and **c)** Nqo13. Sidechains of conserved residues are shown as sticks, water molecules as spheres. **e, g)** Relative ACMA quenching amplitudes for co-reconstituted ATP synthase with **e)** Nqo12^{ATH} and **g)** Nqo13 constructs.

Updated methods:

purification buffer (50 mM HEPES pH 7.5, 200 mM KCl, 150 mM sucrose, 0.005% lauryl maltose neopentyl glycol (LMNG), 20 mM imidazole). The sample was washed using the purification buffer with increasing steps of imidazole concentrations (20 mM, 40 mM, 90 mM). Elution was performed using the purification buffer supplemented with 250 mM imidazole. Relevant fractions were pooled and concentrated using a concentrator with a 100 kDa cut-off. The concentrated sample was loaded in a HiLoad 16/600 Superose 6 pg (Cytiva) for size-exclusion chromatography using SEC buffer (50 mM HEPES pH 7.5, 200 mM KCl, 150 mM sucrose, 0.005% LMNG). Relevant fractions were pooled and concentrated using an Amicon Ultra Protein Concentrator 100K (MWCO). The protein concentration was measured using a NanoDrop spectrophotometer.

Comment/Question:

This is a very impressive body of experimental work and it is neatly documented, starting from expression, purification and proton transport measurements in liposomes.

All three types of measurements described above show the same principal outcome.

- The wt works as proton translocating unit in all three assays
- The glutamic acid residue at the intersubunit bridge seems to control the path, and replacement with glutamine seems to keep the pathway more in the "open" conformation, and the proton transport is accelerated.
- Replacement of a lysine at the exit pathway by an isoleucine seems to decrease the rate of proton transport, potentially by decreasing hydration of the local environment or steric hindrance.

Taken together, I tend to trust in the presented data, that the data show Δ pH driven transport of the individual subunits and that the mutations made affect the transport activity in the proposed fashion. However, it is unsatisfying that the kinetics of the proton transport is not better resolved. Although the determined rates (the unit is not quite clear to me, but I suspect pH units per min), the interpretation is based on the build-up of different steady state levels after a rapid proton transport. The role of second slow kinetic phase, which is well resolved, is not further discussed (but is likely also not relevant).

Answer:

We thank the Reviewer for the excellent and highly encouraging comments. There has indeed been a significant amount of work and development involved in the study, and we very much appreciate that the Reviewer finds our work impressive. To answer the specific points, we now clarify in the revised manuscript that the ATPase rates are in units of " min^{-1} ", while the HPTS-based rate assignments have been removed. Moreover, we clarify that the slow phase is likely to arise from the slow diffusion of protons through the proteoliposome membranes.

Comment/Question:

If the findings are correct, the present manuscript makes an important contribution to solve the current problem of proton transport pathways in complex I, and its data argue strongly against the ND5-only theory by showing that the individual units are able of proton transport. Currently, functional measurements cannot keep up with the speed of new high-resolution structures, and it is important to keep these at a balance. The present study using state-of-the-art MD simulations to make predictions and test them in a well-controlled minimal setup is a brave and powerful attempt to understand the molecular mechanism of proteins like complex. In a very simplified form, an earlier study by Gemperle et al (<https://pubmed.ncbi.nlm.nih.gov/17583799/>) also investigated transport activity of complex I subunits. There, the same subunits (NuoL/N) were expressed in yeast and membrane fraction enriched with the protein were probed for $\text{H}^+/\text{Na}^+/\text{K}^+$ antiport. The present study however goes far beyond that earlier work.

The paper is well written, the introduction is informative and after a very dense simulation section, the experimental work is well described (see below for a few comments).

Answer:

We thank the Reviewer for appreciating our work. Following the Reviewer's suggestion, we have simplified the computational section in the main text to make it more accessible to the general reader. To this end, we have removed detailed technical sections to the SI, for example:

"Several of these potential gating residues are conserved (Supplementary Fig. 18), with conformational changes observed upon proton transfer between the central residues that could regulate the proton uptake and release from the bulk (see Supplementary Fig. 1g, h and Extended discussion)."

Replacing:

"Upon transfer of the proton from Lys235 to Glu377, mimicking a proton transfer step along the central pathway, the sidechain of His292 flips to a different rotameric state, suggesting that the conformation of His292 could affect the back transfer of protons in the reverse direction (cf. also Refs.^{13, 42}, Supplementary Fig. 1g). Moreover, we note that the deprotonation of the Lys235 leads conformational changes in conserved residues in TM7a/b and TM8 (Supplementary Fig. 1h) that could regulate the proton uptake from the N-side bulk."

We have also simplified the discussion of specific residues in the main text, e.g., by simplification of listing specific residues. For example:

"Analogous S-shaped water-mediated proton pathways also form during the MD simulations of the Nqo12^{4TH} construct, with an N-side access pathway (from Glu346, via Lys292 to Lys329); a lateral pathway (from Lys329 via 2-4 water molecules to Lys385); and further a P-side pathway via Asp386 to the P-side bulk (cf. also Refs.^{1, 19, 20, 28, 30}, Fig. 2a, d, Supplementary Fig. 3). Several hydrophilic residues (N-side: Tyr94, Thr23, Ser237, His241; lateral pathway: His321, His325, Gln302, Tyr305) stabilise these proton wires and could support the proton transport (see also Supplementary Fig. 18)."

Question:

The discussion is kept straightforward by arguing that the present data argue against the ND-5 model. They argue that both subunits show similar transport rates, supporting the idea that they might work as a tandem. An interesting question, why mutations in the native complex I favors either 0 or 4 pumped protons (as described in ref. 16) was not touched in the discussion. It would have been interesting how the author relate to the connection between two subunits.

Answer:

The reason for the (0/4) stoichiometry is not known, but we suggest that the coupling between the individual charged elements within and between neighbouring subunits is indeed essential for gating the proton transfer reactions. If the coupling is disturbed, this leads to a proton leak and dissipation of the energy, instead of a pumping step. In terms of our wave propagation model, the perturbed coupling between the elements could hamper the backward wave propagation to the membrane-bound Q site, and an inhibition of the proton pumping machinery. We speculate that severe mutations completely block the wave propagation in both forward and backward directions, whilst softer mutations could allow for the wave

propagation without loss of energy, but may nevertheless also lead to leak at higher PMF conditions.

"The coupling between the individual charged elements within and between neighbouring subunits is essential for gating the proton transfer reactions. If the coupling is disturbed, this leads to a proton leak and dissipation of the energy, instead of a pumping step. In terms of our wave propagation model, the perturbed coupling between the elements could hamper the backward wave propagation to the membrane-bound Q site, and an inhibition of the proton pumping machinery. We speculate that severe mutations completely block the wave propagation in both forward and backward directions at the site of mutation (leading to a pumping stoichiometry of 0¹⁷), whilst softer mutations could allow for the wave propagation without loss of energy, at least in modest PMF conditions (and lead to a pumping stoichiometry of 4¹⁷)."

Question:

Towards the end, the discussion is lead towards the direction of the assembly of the complex and the observed transport rates and I was stumbling on the following sentence:

"In the light of current results, we suggest that these assembly proteins could modulate the rate of proton conduction of the emerging Complex I, and prevent the dissipation of the pmf by blocking the native antiporter contacts, until all components gating the redox-driven proton pumping process are in place." (Beghiah et al., p. 12)."

If isolated and correctly reconstituted subunits nqo12 or 13 were indeed capable of dissipating the pmf, why did it then not happen during the overexpression culture, where these proteins were overexpressed into the cytoplasmic membrane? Maybe a growth test in more challenging media than LB would show such a phenotype?

Answer:

We thank the Reviewer for this excellent suggestion. The growth test in minimal media suggests that the antiporter modules may indeed create a proton leak *in vivo*.

Revisions in the main text and SI below:

"Indeed, the E. coli growth is severely hampered in minimal media (Supplementary Fig. 10m, n), suggesting that the antiporter modules might dissipate part of the pmf during the growth. "

Supplementary Fig. 10 [...]. Growth tests performed in TB and minimum medium (M9) for **m**) Nqo12 and **n**) Nqo13 in induced (I) and non-induced (NI) conditions.

Addition in the Methods section:

"Growth test of the antiporter-like subunits. The expression conditions of WT Nqo12 and Nqo13 were tested as described above both in TB medium and M9 minimum medium. The results of the growth test are shown in Supplementary Fig. 10m,n."

Question:

The mode of dissipation would be $\Delta\psi$ driven proton transport, which might waste the essential potential otherwise used to drive important processes such as nutrient uptake or waste export by secondary active transporters. However, this would imply that the typical mode of action would be $\Delta\psi$ driven transport and not ΔpH driven transport, as the ΔpH is negligible and can even be inverted (e.g. growing *E. coli* at pH 8).

It seems thus crucial to me that the authors connivingly show that the purified subunits are capable to drive pure potential driven proton transport. This has been shown for isolate Fo of ATP synthase (ref. 49) or Na/H antiporters. The details of such experiments are described below.

Transport of Na(+) and K (+) by an antiporter-related subunit from the Escherichia coli NADH dehydrogenase I produced in Saccharomyces cerevisiae - PubMed

The NADH dehydrogenase I from Escherichia coli is a bacterial homolog of the mitochondrial complex I which translocates Na(+) rather than H(+). To elucidate the mechanism of Na(+) transport, the C-terminally truncated NuoL subunit (NuoL(N)) which is related to Na(+)/H(+) antiporters was expressed as ...

<https://pubmed.ncbi.nlm.nih.gov/17583799/>

Answer:

To test whether the antiporter modules support $\Delta\psi$ or ΔpH -driven ion transport, we created external $\Delta\psi$ and ΔpH gradients and induced conduction by addition of valinomycin. Our experiments show that the antiporter modules indeed support both $\Delta\psi$ and ΔpH -driven proton transfer.

Revisions in the main text / SI:

"Although the focus here is on the ΔpH -driven proton transport, we note that the antiporter modules also support a $\Delta\psi$ -driven proton transport, with the valinomycin sensitivity suggesting that the transport could be electrogenic (see Methods, Supplementary Fig. 13b, c, cf. also Ref. 49)."

WT-Nqo12 and WT-Nqo13 mediated proton conduction induced by an external **b)** ΔpH - or **c)** $\Delta\Psi$ - gradient, followed by addition of valinomycin (at 1 min).

Addition to the Methods section:

" ΔpH and $\Delta\Psi$ -mediated ion-transport was performed as described in Ref.⁸⁵ To this end, the ΔpH -mediated proton conduction was tested using 30 μL of proteoliposomes formed in 2 mM MOPS-NaOH pH 7.2, 2.5 mM MgCl_2 , 50 mM KCl, and diluted to 1 mL with 2 mM Bicine-NaOH pH 8.4, 2.5 mM MgCl_2 , 50 mM KCl. The $\Delta\Psi$ -mediated proton conduction was tested by using 30 μL of proteoliposomes formed in 2 mM MOPS-NaOH pH 7.2, 2.5 mM MgCl_2 , 50 mM NaCl, 1 mM KCl, and diluted to 1 mL in 2 mM MOPS-NaOH pH 7.2, 2.5 mM MgCl_2 , 34 mM NaCl, 16 mM KCl. 4 nM valinomycin was added after 1 minute to initiate the proton conduction (see Supplementary Fig. 13b, c)."

85. Fischer S. & Gräber P. Comparison of ΔpH - and $\Delta\phi$ -driven ATP synthesis catalyzed by the H^+ -ATPases from Escherichia coli or chloroplasts reconstituted into liposomes. *FEBS Lett.* 457(3), 327–332 (1999).

Technical questions/comments:

Comments (some of it has been written before the text above and might contain repetitions):

In the following, I list some of my more technical comments on the experiments and suggestions for experiments that might help to alleviate some of the limitations. In my opinion, certain experiments need to be done before the manuscript is ready for publication.

Question:

1. The authors have used valinomycin in all experiments, with identical potassium concentration on either side. The rationale is to relieve a potentially opposing membrane potential that would block proton transport. As a consequence, the observed ΔpH tend to become larger, rates are increased and signals become easier to detect. While this is a correct observation and application, it would be very interesting to see, if the presence of valinomycin and potassium makes an actual difference.

- This would allow to draw direct conclusion if Nqo catalyze electrogenic or electroneutral proton transport. In the native complex I, the transport is electrogenic, and thus omission of valinomycin makes a difference. If however, no difference is observed, isolated Nqo are either distorted and catalyze electroneutral proton transport (e.g. via leaky Cl^- cotransport) or the liposomes are leaky and a stable $\Delta\psi$ cannot be held (see below).

Answer:

We thank Reviewer for this question, which we have addressed above. We now tested the proton conduction with and without valinomycin in the presence of $\Delta\psi$ or ΔpH gradients. As the conduction is indeed sensitive to valinomycin and the antiporter modules support both $\Delta\psi$ and ΔpH -driven proton transport, it supports an electrogenic proton transport.

Question:

2. In all the experiment, a ΔpH is used a driving force. This is somewhat counterintuitive, as the main driving force in e.g. reverse electron transfer is the membrane potential. I therefore strongly suggest driving proton transport by a membrane potential only that can be produced using a potassium valinomycin diffusion potential. The experiment comes with several advantages:

- a.) as $\Delta\psi$ is the driving force, no opposing potential is formed
- b.) no background signal is produced when the reaction is started conveniently by addition of valinomycin
- c.) the direction of transport can be reversed depending on the potassium content of buffer and liposomes.

Answer:

We have addressed this comment above. Our experiments show that the proton conduction can indeed also be achieved with an external $\Delta\psi$. However, to fully address the $\Delta\psi$ vs. ΔpH -driven transport, we would also need to monitor the $\Delta\psi$ components of the PMF (together with a possible exchange of ΔpH to $\Delta\psi$ with, e.g., monensin). Our spectroscopic assays have been currently optimised for the ΔpH -sensitive dye molecules (pyranine and ACMA), while switching to characterisation of $\Delta\psi$ is outside the scope of the present work.

Question:

3. It is unfortunate that the rate of proton transport is not properly resolved. I suggest the following alternatives:

- use a stopped flow apparatus to increase time resolution. The pyranine signal can also just be followed at 460/510 nm, which eliminates the need of using slow ratiometric measurements.

Answer:

This is a good suggestion, but unfortunately outside the scope of the present work. Nevertheless, we have now been able to optimise the time collection resolution to ca. 100 ms, which allowed us to better quantify the kinetics, and accurately identify key differences in the proton conduction properties. We agree that there are certainly interesting fast-kinetics that will be explored in a more detailed study. However, establishing the stopped-flow experiments would require a completely new experimental setup as well as large amount of purified protein, which is currently outside the scope of the present study.

Example of the time-resolution and data used for the kinetic fitting:

Supplementary Fig. 12

Question:

-Instead of adding concentrated base (NaOH) or acid, the liposomes could also be preincubated in an acidic buffer (e.g. 5.5) for some time, and rapidly diluted into buffer at higher pH values. This omits the sometimes-critical step of adding concentrated solutions that might trigger local effects on part of the liposome population. These experiments are well established and are also compatible with stopped-flow measurements, where similar amounts of solutions need to be mixed. See (<https://www.sciencedirect.com/science/article/pii/S0014579399010601> or ref. 49). - decrease the number of Nqo molecules in the liposomes.

Answer:

We agree with the Reviewer that this approach will be a good future development for characterising the fast kinetics. However, we argue that the combined effects of all fast and slow kinetic processes result in the different steady-state conduction levels observed in our experiment. Developing a completely new experimental setup will thus not change the main conclusion of our findings.

Question:

Taking the information from the methods section (200 nm liposomes, 300:1 lipid to protein ratio, Mw~80 kDa, I calculated approximately 10 molecules per liposomes are used. This number can further be decreased, which would slow the transport and making it better detectable. - Increase the buffering capacity of liposomes, e.g. using e.g 20 mM buffer instead of 2 mM buffer

Answer:

We have now better clarified that we used 60 nm liposomes with one antiporter subunit per liposome for the weak acid experiments, and 200 nm liposomes for ATP synthase assays, and we have tested the effect of varying Nqo:liposome ratio (Supplementary Fig. 13). However, due to the time-consuming optimization, we have performed all new experiments in the 2 mM buffer, whilst for the proton transport in the ATPase assays we use a 10 mM buffer concentration.

Revisions in the main text / SI:

“To probe the proton conduction properties of the antiporter modules, we next monitored the proton transfer across proteoliposome membranes using pyranine (8-hydroxypyrene-1,3,6-trisulfonic acid, HPTS) – a fluorescent dye that is quenched as a response of a pH change across the proteoliposome membranes⁴⁴ (see Methods, Fig. 3a, Supplementary Fig. 12a), by optimising the Nqo-to-liposome ratio (Supplementary Fig. 13).”

Supplementary Fig. 13 | Kinetic optimisation and characterisation of proton conduction in proteoliposomes. **a)** Proton conduction monitored in the dissected antiporter-like subunits reconstituted in proteoliposomes with different Nqo-per-liposome ratio. The *pmf* was generated by addition of NaOH. *Inset:* linear relationship between the Nqo:liposome ratio and the resulting steady-state pH level upon addition of base. **b, c)** WT-Nqo12 and WT-Nqo13 mediated proton conduction induced by an external **b)** Δ pH- or **c)** $\Delta\Psi$ - gradient, followed by addition of valinomycin (at 1 min). **d) Left:** ACMA fluorescence quench upon addition of ATP with different Nqo:F₁F_o ratios. *Inset:* linear relationship between the Nqo:liposome ratio and the resulting steady-state pH level upon addition of ATP. *Right:* Closeup of the quenching level.

in Methods:

"A proteoliposome diameter size of 60 nm was used for the acid-induced proton conduction assays, and 200 nm for the ATPase-driven assays (see below), as determined by dynamic light scattering analysis."

Question:

4. As discussed in 1, if the presence of valinomycin makes no difference, this either indicated electroneutral proton transport by Nqo or leaky liposomes. Unfortunately, polar lipid extract, as it has been used in the pyranine measurements, has a bad reputation to make very leaky liposomes (consult Figure S1 from Biochemistry 2012, 51, 8, 1577–1585). In addition, the soybean extract II-s used in the ATP synthase measurements are also considered to produce relatively leaky liposomes. Treatment with ether or acetone might help to remove contaminants to provide charge transfer. Tighter liposomes are typically made from synthetic lipids, either DOPC/E/G or even better POPC/E/G (ref) as the authors have used in their simulation studies.

This is also important in respect to their measurement temperature of 37°C (probably because of of the *Thermus thermophilus* enzyme), which also make liposomes also leakier compared to measurements at room temperature.

Answer:

We appreciate the Reviewer's suggestions. We would like to emphasize that our proteoliposome can withhold the created gradients on the performed timescales as shown by the ATPase assays:

Treatment with acetone or ether might hamper the antiporter proteins, and although we agree that performing the experiments with synthetic lipids could indeed create tighter liposomes, we argue that the polar lipid extract provides a more native-like conditions for the antiporter proteins, as also suggested by our MD simulations (see above). We have therefore decided to keep the current experimental setup.

We have now better clarified in the revised text that:

"During the microsecond molecular dynamics simulations, the isolated antiporter modules remain stable (Supplementary Fig. 1d-f), and dynamically closely resemble the analogous subunits within the intact Complex I simulations (Fig. 1a, b, 2a, b, Supplementary Fig. 1d-f)^{1, 13, 28-31}. Particularly, cardiolipin of the polar lipid model makes tight contacts with the antiporter modules and seals them from the bulk phase (Supplementary Fig. 4)."

"reconstitution of the purified proteins into proteoliposomes comprising E. coli polar lipids (see Methods, Supplementary Fig. 10), to mimic the native bacterial membrane."

"Type II-S soybean lipids were used for preparation of ATP synthase-liposome due to their compatibility with ATP synthase.⁹⁰ To this end, type II-S soybean lipids (ThermoFischer) were dissolved at 5 mg mL⁻¹ into the"

Question:

5. Would it be possible to include a mutant variant that is expected to fully block transport? That would validate the relative findings of the presented mutant variants.

Answer:

By optimizing the proton transport assays, we can now show that three mutations (K235M, K385I and E377Q) indeed fully or partially block the proton conduction relative to the wild-type constructs in the studied conditions. Our MD simulations suggest that these mutations result from the partial drying of the proton conduction pathway.

Revisions in the main text:

"Moreover, our experiments show that proton conduction can also be blocked by substitutions of the key residues along the proton wire in the studied conditions, leading to the same steady-state Δ pH level and proton conduction kinetics as for the empty liposomes. In this regard, we find that the removal of the terminal Lys385 (K385I) in Nqo12^{ΔTH}, or replacement of the 'middle lysine (K235M) or the terminal glutamate (E377Q) in Nqo13 (Fig. 3c) also blocks the proton conduction, suggesting that the residues are important for establishing a tightly gated proton transport machinery in Complex I."

Fig. 3. Biophysical characterisation of proton conduction in dissected antiporter-like modules induced by addition of weak acid. a) Proteoliposome assay for probing the proton conduction kinetics in the dissected antiporter-like subunits with pyranine (HPTS) by addition of K⁺CH₃COO⁻. Addition of 10 mM acetate to b) Nqo12^{ΔTH}- and c) Nqo13- proteoliposomes induced conduction of protons across the membrane. The ionophore nigericin was added to dissipate the generated Δ pH. d, e) The final steady-state Δ pH level before addition of nigericin is shown for the d) Nqo12^{ΔTH} and e) Nqo13 constructs. The protonation conduction rate is sensitive to substitution of the conserved ion-pair and residues along the proton pathway. The data is compared to proton conduction assayed with empty liposomes (EL) and aquaporin (AqpZ) reconstituted proteoliposome. See Supplementary Fig. 10 for addition of weak base, and Supplementary Fig. 13b,c for Δ pH and Δ pH-mediated proton transport.

Predictions from MD simulations on the partial / full drying of the proton channels:

from Supplementary Fig. 2 and 3

Question:

6. As mentioned in 3, the number of Nqo molecules is expected to have a linear impact on the observed signals. In that respect, it is critical to verify that similar amounts of protein is reconstituted into liposomes and is not affected by the mutation.

Answer:

Following the Reviewer's suggestion, we tested the effect of reconstituting increasing amount of Nqo:s per liposome, and found that conduction increases linearly with increasing antiporter:liposome ratio. We have carefully quantified that the same amount of protein is reconstituted into the liposomes (see also answer below).

Revisions in the SI:

Supplementary Fig. 13 | Kinetic optimisation and characterisation of proton conduction in proteoliposomes. **a)** Proton conduction monitored in the dissected antiporter-like subunits reconstituted in proteoliposomes with different Nqo-per-liposome ratio. The *pmf* was generated by addition of NaOH. *Inset:* linear relationship between the Nqo:liposome ratio and the resulting steady-state pH level upon addition of base. **b, c)** WT-Nqo12 and WT-Nqo13 mediated proton conduction induced by an external **b)** Δ pH- or **c)** $\Delta\Psi$ - gradient, followed by addition of valinomycin (at 1 min). **d) Left:** ACMA fluorescence quench upon addition of ATP with different Nqo: F_1F_o ratios. *Inset:* linear relationship between the Nqo:liposome ratio and the resulting steady-state pH level upon addition of ATP. *Right:* Closeup of the quenching level.

Methods:

"prior to addition of the protein sample to the lipids/detergent mixture, which were incubated for 30 min on ice, to incorporate one Nqo protein per vesicle."

Question:

As a worst case scenario, it could be envisaged that replacement of the charged glutamate by a neutral glutamine could have a beneficial effect on membrane protein integration in the absence of its ion salt partner. An increased incorporation would result in increased proton transport as observed, however without necessarily changing the electrostatics of the proton transfer as suggested. On the other hand, I would not expect that protein orientation matters in this case as antiporter like systems often work in either direction

Answer:

We assessed the in-gel liposome fluorescence, suggesting that all constructs show the same amount of incorporation. Moreover, we find that the antiporter modules show a similar orientation in the liposome, while we also agree that the antiporter modules are to conduct in both directions.

Supplementary Fig. 10 [...] o) In-gel fluorescence assessing the reconstitution efficiency of the Nqos into proteoliposomes.

REVIEWER COMMENTS

Reviewer #1 (Remarks to the Author):

The revised version of the work entitled “ Dissected Antiporter Modules Establish Minimal Proton-Conduction Elements in the Respiratory Complex I” presents a significantly improved version of a combined computational, using MD and QM/MM, and experimental study of a minimal proton-conducting membrane modules, created by the authors by engineering and dissecting the key elements of the bacterial Complex I, the highly intricate redox-driven proton pump that powers oxidative phosphorylation across all domains of life.

The authors have done a tremendous amount of additional work to improve the manuscript and adress all my, as well as other reviewer’s, concerns. The work is very well done, figures are impressive and the researchers have a long tradition and experience in working with complex I, which is a relevant topic. The results are reasonably well supported by the presented data. In summary the relevance and quality of the work supports its publication.

In particular and related to suggested the revisions the authors have:

- 1) Reformulated some sections of the manuscript to make it more accessible for the general reader, emphasizing that it addressed the most hotly debated question in the field on whether all antiporter modules of Complex I transport protons across the membrane, or if this activity arises only from the terminal subunit, as recently suggested in two works published in top journals.
- 2) They have re-written parts of the more specialised sections to make them more accessible for the general reader, particularly in relation to the ion-pair hypothesis and how they modulate the proton transfer barrier in the antiporter modules.
- 3) Concerning the methodology, there also seveal key improvements in the current version. They explained that the reaction coordinate for the lateral proton transfer reactions from middle Lys to terminal Lys/Glu, was modelled as a linear combination of bond-breaking and bond-forming distances, currently shown in the Supplementary Figure 6. They also clarified that the focus of these QM/MM free energy simulations is on the lateral proton pathways as these are close to the proposed gating elements and introduced substitutions that they validated experimentally.
- 4) They also calculated using constant pH-MD simulation the pKa of several key residues and placed these pKa shifts in context of our electric wave propagation model, and also discussed how these could be triggered by the Q oxidoreduction activity.

I only have a remaining minor comment, which the authors could consider:

In page 3 the authors mention that “The long-range proton transport process was suggested¹ to take place by an electric wave that propagates in forward and reverse directions across the 200 Å

wide membrane domain, similar to an 'electrical cradle' (cf. Newtonian cradle¹, 13, 37), leading to the release and subsequent uptake of protons by conformational changes in the ion-pairs." Perhaps the authors could just briefly describe what is understood in this context for "electric wave" or "newtonian cradle".

Reviewer #2 (Remarks to the Author):

The authors performed careful work to demonstrate the stability and proper folding of isolated subunits. However, significant unfolding is not necessary to produce artificial H⁺ conduction in liposomes, as H⁺ can move along the boundary of a subunit lacking natural protein contact and exposed to lipids. In this context, the control with aquaporin clearly indicates that the protein, designed to function separately in the membrane, does not affect its permeability properties. The experimental section has been significantly improved. However, in my opinion, it is not correct to directly compare the curve with E123Q13 to other curves, as it started from a different pH value, which is crucial for this type of measurement.

Nevertheless, the data clearly show that reconstituted subunits provide some H⁺ leak. The question arises whether this is an artificial effect or indeed connected to the functioning of H⁺ channels. In the natural environment, these membrane subunits do not transfer protons until they are activated by events following Q reduction. It should also be noted that the operation of complex I is controlled by the transmembrane electrochemical proton potential: at high $\Delta\mu_{\text{H}^+}$, there is neither Q reduction nor H⁺ translocation. This means that under experimental conditions with a high transmembrane pH difference and/or electric potential, the state of membrane channels should be closed. Of course, it is possible that, for some reason, isolated and reconstituted subunits are "stuck" with their channels in the open position. In this case, studies with mutants could shed some light. The mutants from E. coli complex I corresponding to E123Q12, K385I12, E123Q13, and K235I13 almost completely block H⁺ transfer by the enzyme (Refs [14] and [15]); therefore, they should not produce the membrane leak. This is true for K385I12 and K235I13, the other two, on the contrary, increase H⁺ flux (Fig. 3dc).

Moreover, there is some discrepancy between testing proton conduction by acidification with acetate addition (Fig. 3) or ATPase operation (Fig. 4). In the latter figure, K235M13 and E123Q13 did not show significant differences from WT, and only E132Q12 produced a higher proton leak. When proton leak was tested by alkalization (Supplementary Fig. 11), the response of all Nqo12 variants was almost identical (11b), and the same was probably true for Nqo13 variants (11c). However, the curves in this figure are difficult to compare due to different starting points.

It is worth mentioning that replacing only negative residues E123Q13 and E132Q12 resulted in a higher proton flux than the wild type (WT). These residues are situated on the verge of subunits, practically located on the protein/lipid interface upon the reconstitution. In contrast, the positive residues K235M13 and K385I12 are deeper inside the protein body, which could explain their inefficiency in proton transport.

Altogether, the presented data do not unequivocally prove that the observed H⁺ transport was due to the operation of H⁺ channels.

As for the computational study, it would be interesting if the authors compared simulation data on

subunits embedded in lipids and in contact with neighboring membrane subunits. Additionally, it would benefit the paper if the authors summarized the novelty of the modeling.

Reviewer #3 (Remarks to the Author):

The authors have submitted a thoroughly revised version of their manuscript. The new manuscript is improved, easier to read and many of the experiments have been redone, and a better quality of the experimental data is visible facilitating interpretation. Also, some less important measurements have been removed, and an overall improved manuscript is provided.

The data support the proposed model that individual subunits are capable for proton transport and by that support the model in which several and not only nqo12 are involved in pumping. However, the experiments are only indicative, while an exact proof is still lacking, but reasons for this difficulty are discussed.

Despite the lack of ultimate proof of the molecular mechanism, this report nevertheless is a helpful step towards a mechanistic understanding outside of structure interpretation and pure molecular modeling approaches, but instead is a combination of them.

While more experiments, potentially requiring different methods and higher time resolutions, will be necessary to clarify the mechanism in more detail, I am happy to recommend this work for publication in Nature Comm. The much improved language clarity makes it more accessible for scientist outside the direct complex I field.

REVIEWER COMMENTS

Answer to comments by Reviewer #1

The revised version of the work entitled “ Dissected Antiporter Modules Establish Minimal Proton-Conduction Elements in the Respiratory Complex I” presents a significantly improved version of a combined computational, using MD and QM/MM, and experimental study of a minimal proton-conducting membrane modules, created by the authors by engineering and dissecting the key elements of the bacterial Complex I, the highly intricate redox-driven proton pump that powers oxidative phosphorylation across all domains of life.

The authors have done a tremendous amount of additional work to improve the manuscript and address all my, as well as other reviewer's, concerns. The work is very well done, figures are impressive and the researchers have a long tradition and experience in working with complex I, which is a relevant topic. The results are reasonably well supported by the presented data. In summary the relevance and quality of the work supports its publication.

In particular and related to suggested the revisions the authors have:

1) Reformulated some sections of the manuscript to make it more accessible for the general reader, emphasizing that it addressed the most hotly debated question in the field on whether all antiporter modules of Complex I transport protons across the membrane, or if this activity arises only from the terminal subunit, as recently suggested in two works published in top journals.

2) They have re-written parts of the more specialised sections to make them more accessible for the general reader, particularly in relation to the ion-pair hypothesis and how they modulate the proton transfer barrier in the antiporter modules.

3) Concerning the methodology, there also several key improvements in the current version. They explained that the reaction coordinate for the lateral proton transfer reactions from middle Lys to terminal Lys/Glu, was modelled as a linear combination of bond-breaking and bond-forming distances, currently shown in the Supplementary Figure 6. They also clarified that the focus of these QM/MM free energy simulations is on the lateral proton pathways as these are close to the proposed gating elements and introduced substitutions that they validated experimentally.

4) They also calculated using constant pH-MD simulation the pKa of several key residues and placed these pKa shifts in context of our electric wave propagation model, and also discussed how these could be triggered by the Q oxidoreduction activity.

Answer: We thank this Reviewer for their excellent comments and for the suggestions that helped us improve our work. We are very pleased about the positive feedback and appreciation of our work.

I only have a remaining minor comment, which the authors could consider:

In page 3 the authors mention that “The long-range proton transport process was suggested to take place by an electric wave that propagates in forward and reverse directions across the 200 Å wide membrane domain, similar to an 'electrical cradle' (cf. Newtonian cradle¹, 13, 37), leading to the release and subsequent uptake of protons by conformational changes in the ion-pairs.” Perhaps the authors could just briefly describe what is understood in this context for “electric wave” or “newtonian cradle”.

Answer: We thank the Reviewer for the suggestion. We have clarified that the 'Newtonian cradle' refers to a mechanical analogy of the *back-and-forth* charge propagation within the membrane domain. We have removed the unconventional term "electric cradle" to avoid confusion:

"The long-range proton transport process was suggested¹ to take place by an electric wave that propagates in forward and reverse directions across the 200 Å wide membrane domain, in analogy to a Newton's cradle device^{1, 13, 37} (with back-and-forth propagation of mechanical energy), and leading to the release and subsequent uptake of protons by conformational changes in the ion-pairs."

Answer to comments by Reviewer #2

The authors performed careful work to demonstrate the stability and proper folding of isolated subunits. However, significant unfolding is not necessary to produce artificial H⁺ conduction in liposomes, as H⁺ can move along the boundary of a subunit lacking natural protein contact and exposed to lipids. In this context, the control with aquaporin clearly indicates that the protein, designed to function separately in the membrane, does not affect its permeability properties.

The experimental section has been significantly improved. However, in my opinion, it is not correct to directly compare the curve with E123Q13 to other curves, as it started from a different pH value, which is crucial for this type of measurement.

Answer: We thank the Reviewer for appreciating our further validations.

The E123Q¹³ variant mimics a state with an open ion-pair that leads to a faster dissipation of the ΔpH , due to the reduced proton transfer barrier as suggested by our QM/MM simulations. This also implies that in the acid-induced proton conduction experiments, the initial ΔpH gradient starts immediately to equilibrate upon mixing the proteoliposomes in the buffer, thus leading to a higher initial baseline. In this regard, however, the reported final steady-state pH values, shown in Fig. 3d,e, are independent of the baseline, as these show the dissipation of the proton gradient prior to addition of nigericin. Moreover, this initial equilibration phase is not either present in the ATPase assays, where the conduction is induced only upon addition of ATP. The latter assays thus complement the acid-induced conduction experiments, and support the faster proton conduction in the E123Q¹³ variant.

We have now clarified in the revised manuscript that the proton conduction starts upon mixing in the acid-induced experiments, while the final steady-state pH is independent of the initial baseline:

Methods:

“30 μL of PLs were diluted up to 1 mL with proteoliposome buffer, initiating the dissipation of the initial ΔpH across the PL bilayer in presence of 4 nM valinomycin that affect the initial baseline, whereas the final steady-state pH (reported in Fig. 3d,e), obtained prior to addition of nigericin, is independent of the baseline.”

addition to Fig. 3 legend:

“Note that the initial pH gradient pre-equilibrates upon mixing the proteoliposomes in the buffer, leading to a higher initial baseline for the fast-conducting constructs (E123Q¹³ / E132Q¹²), while the final steady-state pH (d,e), obtained prior to addition of nigericin, is independent of the baseline.”

Nevertheless, the data clearly show that reconstituted subunits provide some H⁺ leak. The question arises whether this is an artificial effect or indeed connected to the functioning of H⁺ channels. In the natural environment, these membrane subunits do not transfer protons until they are activated by events following Q reduction. It should also be noted that the operation of complex I is controlled by the transmembrane electrochemical proton potential: at high ΔmH^+ , there is neither Q reduction nor H⁺ translocation. This means that under experimental conditions with a high transmembrane pH difference and/or electric potential, the state of membrane channels should be closed. Of course, it is possible that, for some reason, isolated and reconstituted subunits are "stuck" with their channels in the open position. In this case, studies with mutants could shed some light.

Answer: For the intact Complex I, the proton transport takes place in the reverse direction at a high $\Delta\mu_{\text{H}^+}$, which, in turn, drives the oxidation of quinol (see Introduction of our manuscript):

"Remarkably, this process is fully reversible, and Complex I can thus also operate in the reverse mode, driving quinol oxidation and reverse electron transfer (RET), powered by a Δ pH-gradient across the membrane¹⁸."

We thus expect that the isolated antiporter subunits can also conduct protons in both directions, while the proton conduction is likely to be further controlled by subtle conformational and protonation changes, as supported by our mutagenesis studies (see comments below).

The mutants from *E. coli* complex I corresponding to E123Q12, K385I12, E123Q13, and K235I13 almost completely block H⁺ transfer by the enzyme (Refs [14] and [15]); therefore, they should not produce the membrane leak. This is true for K385I12 and K235I13, the other two, on the contrary, increase H⁺ flux (Fig. 3dc).

Answer: We thank the Reviewer for the suggestion.

We note that it has remained highly challenging to assess the effect of the individual mutations in the full Complex I due to the tight coupling of the protonation and oxidoreductase activities. In this regard, mutation of ion-pair elements or key proton acceptor/donors in the intact Complex I could disrupt the propagation of the "charge wave", and perturb the overall proton-electron coupling (e.g. Refs. 14,15), but without the possibility to determine exactly how the mutations impact the individual proton and electron transfer activities.

To address these challenges, our constructs were designed to probe the effect of functional residues on the pure proton transfer reactions in the individual proton channels. With this approach, we show that ion-pair opening increases the proton conductivity by creating an open state mutant (E123Q¹³, E132Q¹²), while introducing bulky residues along the proton channels (K235M¹³, K385I¹²), partially seals the proton pathway.

We agree that future development is also needed to probe the exact mechanistic principles underlying the proton transport, both in the intact enzyme and in the dissected antiporter-modules. However, our work sheds new light on the regulatory elements of the proton channels. This is now clarified in the main text:

"Mutation of the buried ion-pairs in the intact Complex I results in the complete inhibition of the proton pumping as well as coupled oxidoreductase activity^{12,14,15}, thus hampering systematic studies of how this site modulates the proton transport kinetics. While these studies show that the coupled activity is diminished, they do not allow to determine how the mutations perturb the individual proton transfer reactions within the membrane domain. Here we decoupled the oxidoreduction activity from proton transport, allowing us to resolve how the ion-pair dynamics and individual residue substitutions control the rate of trans-membrane proton transfer."

Moreover, there is some discrepancy between testing proton conduction by acidification with acetate addition (Fig. 3) or ATPase operation (Fig. 4). In the latter figure, K235M13 and E123Q13 did not show significant differences from WT, and only E132Q12 produced a higher proton leak. When proton leak was tested by alkalization (Supplementary Fig. 11), the response of all Nqo12 variants was almost identical (11b), and the same was probably true for Nqo13 variants (11c). However, the curves in this figure are difficult to compare due to different starting points.

Answer: The different assays allow us to probe the proton conduction process in various conditions that complement each other and strengthen our study. While the ATPase conduction assays provide highly controlled experimental conditions, the ACMA dye does not allow us to quantify the exact pH changes, whereas the pyranine dye allows for a quantitative

pH determination, but in less controlled conditions. While the two approaches are highly complementary, the experiments require different buffer conditions and preparation, as described in the methods section.

The conduction induced by alkalization is more difficult to interpret due to the high pK_a of 10.6 of methylamine, which results in a smaller optical response. These data, nevertheless, complement the acid-induced experiments, but are not as conclusive. This is stated in the legend of Supplementary Fig. 11:

"Methylamine ($pK_a=10.6$) addition leads to smaller kinetic differences between the constructs as compared to the acetate ($pK_a=4.8$) assays, possibly due to the shifted equilibrium between the acid and base forms ($[acid]/[base] = 10^{3.4}$ amine vs. $10^{2.4}$ acetate at pH 7.2), secondary reactions between hydroxide ions and $CH_3NH_3^+$, and/or differences in membrane permeability of the weak acid/base ($P_{CH_3COOH} = 6.9 \times 10^{-3} \text{ cm s}^{-1}$, $P_{CH_3NH_2} = 9 \times 10^{-1} - 8 \times 10^{-2} \text{ cm s}^{-1}$)."

It is worth mentioning that replacing only negative residues E123Q13 and E132Q12 resulted in a higher proton flux than the wild type (WT). These residues are situated on the verge of subunits, practically located on the protein/lipid interface upon the reconstitution. In contrast, the positive residues K235M13 and K385I12 are deeper inside the protein body, which could explain their inefficiency in proton transport.

Altogether, the presented data do not unequivocally prove that the observed H^+ transport was due to the operation of H^+ channels.

Answer: We thank the Reviewer for the suggestion that the buried residues might prevent their ability to transport protons. In this regard, our simulations suggest the proton pathways are connected via K235¹³ and K385¹², and the residues undergo protonation changes, whereas the K235M¹³ and K385I¹² substitutions block the proton wires. Additionally, our simulations suggest that the E123Q¹³ and E132Q¹² variants induce a conformational change in K204¹³ and K216¹² that lowers the proton transfer barrier, whereas the E377Q¹³ substitution, also located near the protein/lipid interface, lowers the proton conductivity, and further supports the role of these residues in the proton conduction. Taken together, our combined findings suggest that all of these residues are highly conserved and central for the proton transport activity. This is described in the main text:

*"In this regard, the steady-state ΔpH level reaches 7.1-7.12 for the WT Nqo13 and Nqo12^{4TH} relative to the empty liposomes/AqpZ ($\Delta pH < 7.04$), while the **substitution of the ion-pair glutamate** at the position Glu123 (E123Q-Nqo13) or Glu132 (E132Q-Nqo12) strongly increases the proton conduction rates (Fig. 3b, c, Supplementary Table 1), and leading to a further shift in the steady-state ΔpH level to a pH of around 7.15, consistent with our computational predictions of the lowered proton transfer barrier (Fig. 2c-j). Moreover, our experiments show that proton conduction can also be blocked by substitutions of the key residues along the proton wire in the studied conditions, leading to the same steady-state ΔpH level and proton conduction kinetics as for the empty liposomes. In this regard, we find that the removal of the terminal Lys385 (K385I) in Nqo12^{4TH}, or replacement of the 'middle lysine (K235M) or the terminal glutamate (E377Q) in Nqo13 (Fig. 3c) also blocks the proton conduction, suggesting that the residues are important for establishing a tightly gated proton transport machinery in Complex I."*

We agree with the Reviewer that it is difficult to unequivocally prove the mechanism of proton conduction in the proton channel. However, we argue that our combined findings strongly support the role of these residues in the proton transport.

As for the computational study, it would be interesting if the authors compared simulation data on subunits embedded in lipids and in contact with neighboring membrane subunits.

Answer: We thank the Reviewer for this excellent suggestion. We now show that MD simulations of the intact Complex I show similar hydration in the protein/lipid interface as in the dissected antiporter modules. We have included a new SI figure showing the hydration analysis at the protein/lipid interface in the individual constructs and in simulations of full Complex I:

“Nqo12 lacks a neighbouring subunit on one side, and we find overall smaller pK_a shifts upon isolation of Nqo12 (Supplementary Fig. 15, Supplementary Table 3), suggesting that its protonation dynamics could resemble that of the intact Complex I. In this regard, our constructs also show similar hydration at the protein-lipid interface as compared to the protein-protein interface of neighbouring subunits in simulations of the intact Complex I⁰ (Supplementary Fig. 19).”

Supplementary Fig. 19 | Hydration analysis of the protein-lipid interface in the isolated antiporter constructs, and in the full Complex I. Left insets show a snapshot of the hydration ensemble with water molecules within 3 Å of protein and lipids. Right insets show the distribution of water molecules along the Z axis. The analyses were performed on the last 500 ns of the respective MD simulation, with structures extracted every 10 ns. **A-D)** MD simulations of the Nqo13 constructs. **A)** WT, **B)** E123Q, **C)** K235M, and **D)** E377Q. **C-G)** MD simulations of the Nqo12^{ATM} constructs. **E)** WT, **F)** E132Q, and **G)** K385I. MD simulations of the full Complex I, showing water molecules around the protein-lipid interface at the Nqo12/Nqo13 subunits (based on simulation data from Ref. 7).

Additionally, it would benefit the paper if the authors summarized the novelty of the modeling.

Answer: Our work shows that the antiporter subunits comprise the minimal proton conducting units in Complex I, and it demonstrates the molecular principles that control the proton conduction process. The experimentally studied systems were first designed and tested computationally, followed by our detailed experimental characterization of the constructs. As these engineered constructs are novel, they have not been studied before. Our simulations provide important functional insight into the proton conduction mechanism, including their underlying electrostatic gating principles that are difficult to assess experimentally. However, we argue that the integration of computational and experimental data provides here a unique structural and mechanistic insight into the proton transfer process. We believe that this message is well summarized in the manuscript:

"To derive a bottom-up understanding of the intricate proton pumping process, we design here minimal proton-conducting modules of Complex I that allow us for the first time to probe the molecular principles underlying the putative 'conductive' states during turnover within the smallest proton-conducting unit of Complex I. Our constructs are built based on multi-scale simulations, and probed experimentally by biophysical proton conduction experiments in proteoliposomes... "

"To rationally engineer and dissect the pumping machinery of Complex I, we first created atomistic molecular dynamics (MD) simulation models of the antiporter modules"

"We have shown here using a multi-layered experimental and computational approach that the isolated antiporter modules of Complex I contain all necessary elements to transport protons across proteoliposome membranes. We further found that the rate of proton conduction is controlled by the conformational state of a buried ion-pair, with electric field effects and wetting transitions forming a basis for the gating mechanism. Mechanistic studies of the isolated antiporter modules could provide central understanding of how key residues involved in, e.g., mitochondrial diseases affect the elementary proton transport properties, which are difficult to assess in the native intact Complex I. Taken together, our findings show how the antiporter modules and their interactions gate the proton pumping in Complex I, highlighting key functional principles of its unresolved long-range energy transduction mechanism."

Reviewer #3

The authors have submitted a thoroughly revised version of their manuscript. The new manuscript is improved, easier to read and many of the experiments have been redone, and a better quality of the experimental data is visible facilitating interpretation. Also, some less important measurements have been removed, and an overall improved manuscript is provided.

The data support the proposed model that individual subunits are capable for proton transport and by that support the model in which several and not only nqo12 are involved in pumping. However, the experiments are only indicative, while an exact proof is still lacking, but reasons for this difficulty are discussed.

Despite the lack of ultimate proof of the molecular mechanism, this report nevertheless is a helpful step towards a mechanistic understanding outside of structure interpretation and pure molecular modeling approaches, but instead is a combination of them.

While more experiments, potentially requiring different methods and higher time resolutions, will be necessary to clarify the mechanism in more detail, I am happy to recommend this work for publication in Nature Comm. The much improved language clarity makes it more accessible for scientist outside the direct complex I field.

Answer: We are very pleased that the Reviewer recommends our work for publication in Nature Communications, and thank them for their suggestions. Future detailed mechanistic studies are indeed in progress.

REVIEWERS' COMMENTS

Reviewer #2 (Remarks to the Author):

The authors studied reconstituted Nuo13 and Nuo12 subunits of complex I, focusing on the replacement of amino acid residues known to be essential for proton pumping. The paper claims that the data provide direct experimental evidence that the individual antiporter modules are responsible for the proton transport activity of Complex I. However, I am not convinced that there is direct evidence. Firstly, only one mutation, E123Q12, out of five resulted in a clear increase in proton conductivity. The others showed responses either similar to the wild type (WT), such as K385I12 and E123Q13, or dependently on conditions resulted in some decrease in proton conductivity, like K235M13 and E377Q13, which is not particularly informative. Secondly, the main question of whether these mutations produce an artificial effect by disturbing the bilayer with improperly folded or aggregated proteins, or whether they are indeed connected to the functioning of H⁺ channels in the natural environment, is not addressed. The juxtaposition of experimental and simulation data hardly can resolve this question.

It is difficult to agree that “ If the coupling is disturbed, this leads to a proton leak and dissipation of the energy, instead of a pumping step”. Proton leak from non-operating Complex I is highly unfavorable physiologically and never observed experimentally. In favor of such leak the authors make the argument that E.coli cells with replacement of essential for proton transport residues do not grow on minimal medium due to proton leak. However, it depends on the medium content. If the medium is supplemented with particular substrate, e.g. malate or lactate, Complex I is the main source of energy supply, so cells cannot grow without functioning Complex I. At the same time, they grow perfectly in the rich medium.

Although the data on reconstituted Nuo13 and Nuo12 subunits are not conclusive, this study represents, to my knowledge, the second attempt after Steuber J. (*J Biol Chem.* 2003 Jul 18;278(29):26817-22) to evaluate the properties of isolated membrane subunits of complex I. It could serve as a good basis for future work.

REVIEWERS' COMMENTS

Reviewer #2 (Remarks to the Author):

Answer: We thank this Reviewer for their constructive comments in the previous revision round that have helped us improve our work. The additional comments are addressed below.

The authors studied reconstituted Nuo13 and Nuo12 subunits of complex I, focusing on the replacement of amino acid residues known to be essential for proton pumping. The paper claims that the data provide direct experimental evidence that the individual antiporter modules are responsible for the proton transport activity of Complex I. However, I am not convinced that there is direct evidence. Firstly, only one mutation, E123Q12, out of five resulted in a clear increase in proton conductivity. The others showed responses either similar to the wild type (WT), such as K385I12 and E123Q13, or dependently on conditions resulted in some decrease in proton conductivity, like K235M13 and E377Q13, which is not particularly informative.

Answer: We thank the Reviewer for these additional comments. We respectfully disagree on assessment that our constructs are not informative or that nearly all constructs resulted in the same conduction level. Our data unambiguously show that 1) the dissected WT-Nqo12 and WT-Nqo13 are able to conduct protons and 2) that the ion-pair mutants (E132Q¹² and E123Q¹³) have a statistically significant increase of the proton conduction in both ATP synthase assays and by acid addition (Fig. 3d, e, and Fig. 4d, f). In contrast, the other mutations (K385I¹², K235M¹³ and E377Q¹³) showed partial or full blockage of the proton conduction in both experiments (Figs. 3 and 4).

The mutagenesis work was done to study residues with a targeted purpose. While the ion-pair mutations were created to mimic a conformation that results in a lowered lateral proton conduction barrier (see Fig. 2), the other mutants were designed to block the conduction of protons, as indicated by the reduced hydration along the proton channel (Fig. 2, Supplementary Fig. 2, and 3).

Secondly, the main question of whether these mutations produce an artificial effect by disturbing the bilayer with improperly folded or aggregated proteins, or whether they are indeed connected to the functioning of H⁺ channels in the natural environment, is not addressed. The juxtaposition of experimental and simulation data hardly can resolve this question.

Answer: We have further clarified that it would not be possible to block the proton conduction by mutations within the antiporter constructs, if the transport activity would result as an artifact of protein unfolding or aggregation. Specifically, the K385I¹², E377Q¹³ and K235M¹³ constructs show a similar proton conduction as the aquaporin and the empty proteoliposomes (Fig. 3d, e). As also emphasized in the main text, these mutations lead to

a local drying of the proton channels in the MD simulations (Fig. 2, Supplementary Figs. 2, 3), further providing a molecular basis for the effect. The proper folding of our protein constructs is supported by analysis of our SEC profiles, GFP-FSEC assays (Supplementary Fig. 10d, j, k), and in-gel fluorescence of proteoliposomes (Supplementary Fig. 10e). It is thus highly unlikely that the described proton conduction is an artificial effect that disturbs the bilayer by improperly folded or aggregated proteins.

These aspects have now been further clarified in the main text:

"These differences are highly unlikely to arise from protein aggregation or unfolding effects, as shown by size-exclusion chromatography and FSEC-GFP profiles (Supplementary Fig. 10d, j, k) and in-gel fluorescence (Supplementary Fig. 10o). Moreover, aggregated protein constructs would not allow us to both enhance and block the proton conduction to the control level by introducing rational mutations (see below)."

"Moreover, our dissected protein constructs show no signs of unfolding or aggregation (Supplementary Figs. 10) that could lead to artificial proton leaks."

It is difficult to agree that " If the coupling is disturbed, this leads to a proton leak and dissipation of the energy, instead of a pumping step". Proton leak from non-operating Complex I is highly unfavorable physiologically and never observed experimentally.

Answer: We respectfully disagree with this statement. Mutations that disrupt the coupling certainly disturb the PCET reactions in Complex I, as observed in many experiments (refs 10-17), where both the oxidoreduction and proton pumping activities are drastically reduced. Moreover, there are several examples of disease-related mutations in Complex I, which also disturb the function of Complex I (and most likely the PCET coupling) that cause severe physiological effects.

Addition in the main text:

If the coupling is disturbed, this **could lead** to a proton leak and dissipation of the energy, instead of a pumping step, **as suggested by inhibition of the coupled proton pumping and oxidoreductase activity**¹⁰⁻¹⁷.

In favor of such leak the authors make the argument that E.coli cells with replacement of essential for proton transport residues do not grow on minimal medium due to proton leak. However, it depends on the medium content. If the medium is supplemented with particular substrate, e.g. malate or lactate, Complex I is the main source of energy supply, so cells cannot grow without functioning Complex I. At the same time, they grow perfectly in the rich medium.

Answer: These assays were conducted to address a suggestion by another Reviewer. As the isolated WT constructs have all the necessary elements needed to conduct protons, it begs the question of what prevents proton leaks during assembly of the intact Complex I. However, as the conduction is relatively slow on a cellular scale, it would not affect normal

growth unless the expression is performed under more strained minimal medium conditions. To this end, we tested the expression under normal as well as highly strained minimal medium conditions, which indeed showed an effect. However, we do not claim that these findings would have direct implications *in vivo*.

We appreciate the Reviewer's suggestion on exploring the expression by supplementing the medium additionally with different substrates, but it would be unlikely that such experiments would reveal additional effects beyond those already shown in the minimal medium conditions.

Additions to the main text:

However, it remains important that the **individual** antiporter modules do not dissipate the *pmf* that could hamper the energy metabolism. Indeed, the *E. coli* growth is severely hampered in minimal media (Supplementary Fig. 10m, n), suggesting that the antiporter modules might dissipate part of the *pmf* during the growth, **although it is difficult to draw detailed conclusions of the dissipation effects *in vivo* conditions based on these observations. We note that the...**

Although the data on reconstituted Nuo13 and Nuo12 subunits are not conclusive, this study represents, to my knowledge, the second attempt after Steuber J. (J Biol Chem. 2003 Jul 18;278(29):26817-22) to evaluate the properties of isolated membrane subunits of complex I. It could serve as a good basis for future work.

Answer: The early work by Steuber explored the Na⁺ conduction in *E. coli* NuoL. Based on this, it was suggested that the NuoL domain pumps two sodium ions, which contrasts the modern knowledge of Complex I function. Although we cannot comment on the details of this study, it is possible that Na⁺ leakage could have been generated by protein impurities and/or unfolded proteins in vesicles. Indeed, our early attempts to express the dissected antiporter constructs in non-optimal vectors and strains, created low yields and high impurities, which were difficult to quantify in proteoliposome assays. In contrast, our protein models presented in this work can be biochemically expressed, purified, and quantitatively controlled (as shown in Supplementary Fig. 10, Supplementary Table 7) and they do not leak Na⁺ ions (Supplementary Fig. 13c).